# SINC KOLMOGOROV-ARNOLD NETWORK AND ITS APPLICATION FOR FUNCTIONS WITH SINGULARITIES

## ABSTRACT

In this paper, we propose to use Sinc interpolation in the context of Kolmogorov-Arnold Networks, neural networks with learnable activation functions, which recently gained attention as alternatives to multilayer perceptron. Many different function representations have already been tried, but we show that Sinc interpolation proposes a viable alternative, since it is known in numerical analysis to represent well both smooth functions and functions with singularities. This is important not only for function approximation but also for the solutions of partial differential equations with physics-informed neural networks. Through a series of experiments, we show that SincKANs provide better results in almost all of the examples we have considered.

## 1 INTRODUCTION

Multilayer perceptron (MLP) is a classical neural network consisting of fully connected layers with a chosen nonlinear activation function, which is a superposition of simple functions. The classical Kolmogorov-Arnold representation theorem Kolmogorov (1961); Arnol'd (1959) states that every function can be represented as a superposition of function of at most 2 variables, motivating the research for learnable activation functions.

Kolmogorov's Spline Network (KSN) Igelnik & Parikh (2003) is a two-layer framework using splines as the learnable activation functions. Recently, Kolmogorov-Arnold Networks (KANs) Liu et al. (2024b) sparkled a new wave of attention to those approaches, by proposing a multilayer variant of KSN. Basically, any successful basis to represent univariate functions can provide a new variant of KAN. Many well-known methods have already been investigated including wavelet Bozorgasl & Chen (2024); Seydi (2024b), Fourier series Xu et al. (2024), finite basis Howard et al. (2024), Jacobi basis functions Aghaei (2024a), polynomial basis functions Seydi (2024a), rational functions Aghaei (2024b) and Chebyshev polynomials SS (2024); Shukla et al. (2024).

We propose to use Sinc interpolation (the Sinc function is defined in Eq. (4)) which is a very efficient and well-studied method for function interpolation, especially 1D problems Stenger (2016). To our knowledge, it has not been studied in the context of KANs. We argue that the the cubic spline interpolation used in KANs should be replaced by the Sinc interpolation, because splines are particularly good for the approximation of analytic functions without singularities which MLP is also good at, while Sinc methods excel for problems with singularities, for boundary-layer problems, and for problems over infinite or semi-infinite range Stenger (2012). Herein, utilizing Sinc functions can improve the accuracy and generalization of KANs, and make KANs distinguishing and competitive, especially in solving aforementioned mathematical problems in machine learning. We will confirm our hypothesis by numerical experiments.

Physics-informed neural networks (PINNs) Lagaris et al. (1998); Raissi et al. (2019) are a method used to solve partial differential equations (PDEs) by integrating physical laws with neural networks in machine learning. The use of Kolmogorov-Arnold Networks (KANs) in PINNs has been explored and is referred to as Physics-Informed Kolmogorov-Arnold Networks (PIKANs) Rigas et al. (2024); Wang et al. (2024). Due to the high similarity between KAN and MLP, PIKANs inherit several advantages of PINNs, such as overcoming the curse of dimensionality (CoD) Wojtowytsch & Weinan (2020); Han et al. (2018), handling imperfect data Karniadakis et al. (2021), and performing interpolation Sliwinski & Rigas (2023). PINNs have diverse applications, including fluid dynamics Raissi et al. (2020); Jin et al. (2021); Kashefi & Mukerji (2022), quantum mechanical systems Jin

et al. (2022), surface physics Fang & Zhan (2019), electric power systems Nellikkath & Chatzivasileiadis (2022), and biological systems Yazdani et al. (2020). However, they also face challenges such as spectral bias Xu et al. (2019); Wang et al. (2022), error estimation Fanaskov et al. (2024), and scalability issues Yao et al. (2023).

In this paper, we introduce a novel network architecture called Sinc Kolmogorov-Arnold Networks (SincKANs). This approach leverages Sinc interpolation, which is particularly adept at approximating functions with singularities, to replace cubic interpolation in the learnable activation functions of KANs. The ability to handle singularities enables SincKAN to mitigate the spectral bias observed in PIKANs, thereby making PIKANs more robust and capable of solving PDEs that traditional PINNs may struggle with. Additionally, we conducted a series of experiments to validate SincKAN's interpolation capabilities and assess their performance as a replacement for MLP and KANs in PINNs. Our specific contributions can be summarized as follows:

1. We propose the Sinc Kolmogorov-Arnold Networks, a novel network that excels in handling singularities.

2. We propose several approaches based on classical techniques of Sinc methods that can enhance the robustness and performance of SincKAN.

3. We conducted a series of experiments to demonstrate the performance of SincKAN in approximating a function and PIKANs.

The paper is structured as follows: In Section 2, we briefly introduce the PINNs, discuss Sinc numerical methods, and provide a detailed explanation of SincKAN. In Section 3 we compare our SincKAN with several networks including MLP, modified MLP Wang et al. (2021), KAN, ChebyKAN in several diverse benchmarks including smooth functions, discontinuous functions, and boundary layer problems. In Section 4, we conclude the paper and discuss the remaining limitations and directions for future research.

## 2 METHODS

### 2.1 PHYSICS-INFORMED NEURAL NETWORKS (PINNS)

We briefly review the physics-informed neural networks (PINNs) Raissi et al. (2019) in the context of inferring the solutions of PDEs. Generally, we consider time-dependent PDEs for $\boldsymbol{u}$ taking the form

$$
\begin{aligned}
\partial_t \boldsymbol{u} + \mathcal{N}[\boldsymbol{u}] &= 0, \quad t \in [0, T], \ \boldsymbol{x} \in \Omega, \\
\boldsymbol{u}(0, \boldsymbol{x}) &= \boldsymbol{g}(\boldsymbol{x}), \quad \boldsymbol{x} \in \Omega, \\
\mathcal{B}[\boldsymbol{u}] &= 0, \quad t \in [0, T], \ \boldsymbol{x} \in \partial\Omega,
\end{aligned}
\tag{1}
$$

where $\mathcal{N}$ is the differential operator, $\Omega$ is the domain of grid points, and $\mathcal{B}$ is the boundary operator. When considering time-independent PDEs, $\partial_t \boldsymbol{u} \equiv 0$.

The ambition of PINNs is to approximate the unknown solution $\boldsymbol{u}$ to the PDE system Eq. (1), by optimizing a neural network $\boldsymbol{u}^\theta$, where $\theta$ denotes the trainable parameters of the neural network. The constructed loss function is:

$$
\mathcal{L}(\theta) = \mathcal{L}_{ic}(\theta) + \mathcal{L}_{bc}(\theta) + \mathcal{L}_r(\theta),
\tag{2}
$$

where

$$
\mathcal{L}_r(\theta) = \frac{1}{N_r} \sum_{i=1}^{N_r} \left| \partial_t \boldsymbol{u}^\theta \left( t_r^i, \boldsymbol{x}_r^i \right) + \mathcal{N} \left[ \boldsymbol{u}^\theta \right] \left( t_r^i, \boldsymbol{x}_r^i \right) \right|^2,
$$

$$
\mathcal{L}_{ic}(\theta) = \frac{1}{N_{ic}} \sum_{i=1}^{N_{ic}} \left| \boldsymbol{u}^\theta \left( 0, \boldsymbol{x}_{ic}^i \right) - \boldsymbol{g} \left( \boldsymbol{x}_{ic}^i \right) \right|^2,
\tag{3}
$$

$$
\mathcal{L}_{bc}(\theta) = \frac{1}{N_{bc}} \sum_{i=1}^{N_{bc}} \left| \mathcal{B} \left[ \boldsymbol{u}^\theta \right] \left( t_{bc}^i, \boldsymbol{x}_{bc}^i \right) \right|^2,
$$

corresponds to the three equations in Eq. (1) individually; $\boldsymbol{x}_{ic}^i, \boldsymbol{x}_{bc}^i, \boldsymbol{x}_r^i$ are the sampled points from the initial constraint, boundary constraint, and residual constraint, respectively; $N_{ic}, N_{bc}, N_r$ are

the total number of sampled points for each constraint, correspondingly. Note that in Raissi et al. (2019), $u^\theta(x) = \mathbf{MLP}(x)$.

## 2.2 SINC NUMERICAL METHODS

The Sinc function is defined as[1]

$$\text{Sinc}(x) = \frac{\sin(x)}{x}, \tag{4}$$

the Sinc series $S(j,h)(x)$ used in Sinc numerical methods is defined by:

$$S(j,h)(x) = \frac{\sin[(\pi/h)(x-jh)]}{(\pi/h)(x-jh)}, \tag{5}$$

then the Sinc approximation for a function $f$ defined on the real line $\mathbb{R}$ is given by

$$f(x) \approx \sum_{j=-N}^{N} f(jh)S(j,h)(x), \quad x \in \mathbb{R}, \tag{6}$$

where $h$ is the step size with the optimal value $\sqrt{\pi d/\beta N}$ provided in Theorem 1, and $2N+1$ is the degree of Sinc series.

Thanks to Sinc function's beautiful properties including the equivalence of semidiscrete Fourier transform Trefethen (2000), its approximation as a nascent delta function, etc., Sinc numerical methods have become a technique for solving a wide range of linear and nonlinear problems arising from scientific and engineering applications including heat transfer Lippke (1991), fluid mechanics Abdella (2015), and solid mechanics Abdella et al. (2009). But Sinc series are the orthogonal basis defined on $(-\infty, \infty)$ which is impractical for numerical methods. To use Sinc numerical methods, one should choose a proper coordinate transformation based on the computing domain $(a,b)$ and an optimal step size based on the target function $f$. However, manually changing the network to meet every specific problem is impractical and wasteful. In the following of this section, we will introduce current techniques used in Sinc numerical methods. Then in Section 2.3, we will unfold Sinc numerical methods to meet machine learning. At first, we introduce the convergence theorem:

**Theorem 1.** *Sugihara & Matsuo (2004)*

*Assume $\alpha, \beta, d > 0$, that*

*(1) $f$ belongs to $H^1(\mathcal{D}_d)$, where $H^1$ is the Hardy space and $\mathcal{D}_d = \{z \in \mathbb{C} \mid |\Im z| < d\}$;*

*(2) $f$ decays exponentially on the real line, that is, $|f(x)| \leq \alpha \exp(-\beta|x|)$, $\forall x \in \mathbb{R}$.*

*Then we have*

$$\sup_{-\infty < x < \infty} \left| f(x) - \sum_{j=-N}^{N} f(jh)S(j,h)(x) \right| \leq CN^{1/2} \exp\left[ -(\pi d\beta N)^{1/2} \right] \tag{7}$$

*for some constant $C$, where the step size $h$ is taken as*

$$h = \left( \frac{\pi d}{\beta N} \right)^{1/2}. \tag{8}$$

Theorem 1 indicates that the exponential convergence of Sinc approximation on the real line depends on the parameters $d, \beta$, determined by the target function $f$. Thus, in Sinc numerical methods, researchers set specific parameters for specific function $f$ Sugihara & Matsuo (2004); Mohsen (2017) or set them by bisection Richardson & Trefethen (2011). Note that both approaches require the target function $f$, but in machine learning, $f$ is usually unknown.

In numerical mathematics, practical problems generally require approximating on an interval $(a,b)$ instead of the entire real line $\mathbb{R}$. To implement Sinc methods on general functions, we have to

---

[1]In engineering, they define Sinc function as $\text{Sinc}(x) = \frac{\sin(\pi x)}{\pi x}$

transform the interval $(a, b)$ to $\mathbb{R}$ with a properly selected coordinate transformation, *i.e.* we define a transformation $x = \psi(\xi)$ such that $\psi : (-\infty, +\infty) \to (a, b)$. Then Eq. (6) is replaced by

$$f(\psi(\xi)) \approx \sum_{j=-N}^{N} f(\psi(jh))S(j, h)(\xi), \quad -\infty < \xi < \infty, \tag{9}$$

where $h$ is the step size with the optimal value $\sqrt{\pi d'/\beta' N}$ provided in Theorem 2. The following theorem states that Theorem 1 still holds with some different $\alpha, \beta$, and $d$ after the coordinate transformation.

**Theorem 2.** *Sugihara & Matsuo (2004)*

*Assume that, for a variable transformation $x = \psi(\xi)$, the transformed function $f(\psi(\xi))$ satisfies assumptions 1 and 2 in Theorem 1 with some $\alpha', \beta'$ and $d'$. Then we have*

$$\sup_{a \leq x \leq b} \left| f(x) - \sum_{j=-N}^{N} f(\psi(jh))S(j, h)\left(\psi^{-1}(x)\right) \right| \leq CN^{1/2} \exp\left[-(\pi d'\beta' N)^{1/2}\right]$$

*for some $C$, where the step size $h$ is taken as $h = \sqrt{\frac{\pi d'}{\beta' N}}$*

This theorem suggests the possibility that even a function $f$ with an end-point singularity can be approximated successfully by Eq. (9) with a suitable choice of transformation.

Furthermore, we empirically demonstrate the merits of Sinc methods in Fig. 1 via numerical results generated by Chebfun Driscoll et al. (2014) for Chebyshev and cubic spline interpolation and Sincfun Richardson & Trefethen (2011) for Sinc interpolation.

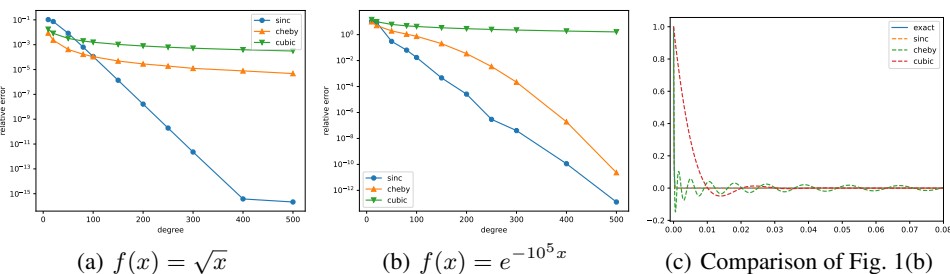

(a) $f(x) = \sqrt{x}$      (b) $f(x) = e^{-10^5 x}$      (c) Comparison of Fig. 1(b)

Figure 1: Fig. 1(a) depicts the Sinc's merit of handling the end-point singularity while the Chebyshev and the spline converge slowly. Fig. 1(b) shows that, for the boundary layer functions that have high derivatives, Sinc converges exponentially while Chebyshev converges slowly at first. Fig. 1(c) partially depicts the solution and the interpolations over the interval $[0, 0.08]$, indicating that Sinc interpolation provides the most accurate approximation, while Chebyshev interpolation exhibits significant oscillations, and spline interpolation shows localized inaccuracies in certain regions.

### 2.3 SINC KOLMOGOROV-ARNOLD NETWORK (SINCKAN)

Suppose $\mathbf{\Phi} = \{\phi_{p,q}\}$ is the matrix of univariate functions where $p = 1, 2, \dots n_{in}, q = 1, 2, \dots n_{out}$. Then the L-layers Kolmogorov-Arnold Networks can be defined by[2]:

$$\text{KAN}(\boldsymbol{x}) = (\mathbf{\Phi}_{L-1} \circ \mathbf{\Phi}_{L-2} \circ \cdots \circ \mathbf{\Phi}_1 \circ \mathbf{\Phi}_0) \boldsymbol{x}, \quad \boldsymbol{x} \in \mathbb{R}^d. \tag{10}$$

In vanilla KAN Liu et al. (2024b), every univariate function $\phi$ is approximated via a summation with cubic spline:

$$\phi_{\text{spline}}(x) = w_b \text{silu}(x) + w_s \left( \sum_i c_i B_i(x) \right), \tag{11}$$

---

[2]the detailed explanation of KAN is in Appendix G.3

where $c_i, w_b, w_s$ are trainable parameters, and $B_i$ is the spline. Intuitively, to replace cubic interpolation with Sinc, we can define:

$$\phi_{\text{single}}(x) = \sum_{i=-N}^{N} c_i S(i, h)(x), \tag{12}$$

where $c_i$ is trainable parameters, if $h$ is set to the optimal value Eq. (8), the optimal approximation of $f(x)$ in Eq. (6) is $c_i^* = f(ih), \forall i = -N, \cdots, N$. However, to replace the interpolation method successfully, the aforementioned techniques require further investigations:

**Optimal $h$**   As we discussed in Section 2.2, it is impractical to set a single optimal $h$ in machine learning frameworks. Thus in SincKAN, we propose an extension to the Sinc approximation with a mixture of different step sizes $h_j$:

$$\phi_{\text{multi}}(x) = \sum_{j=1}^{M} \sum_{i=-N}^{N} c_{i,j} S(i, h_j)(x), \tag{13}$$

where $c_{i,j}$ are trainable parameters. Stenger (2012) states that, if the chosen $h$ is larger than the optimal value predicted by Eq. (8), the interpolation is less accurate near the origin and more accurate farther away from the origin; if the chosen $h$ is smaller than the optimal value predicted by Eq. (8), the interpolation is more accurate near the origin and less accurate farther away from the origin. Herein, combining different $h$ with adaptive weights can result in a more accurate approximation than the optimal $h$ and doesn't need to calculate the optimal $h$. Thus, compared to Eq. (12), expanding the approximation by a summation of several different $h_j$ can not only avoid determining $h$ for every specific function but also improve the accuracy.

**Coordinate transformation**   Another challenge is the choice of coordinate transformation which is also problem-specific Stenger (2000). Let's inherent the notation of Section 2.2, and suppose $\mathcal{X} = \{x_i\}_{i=1}^{N}$ is the ordered set of input points with $x_1 \leq x_2 \leq \cdots \leq x_N$, then we can define the open interval $(a, b)$ by $a = x_1 - \epsilon, b = x_N + \epsilon$, where $\epsilon$ is a chosen number, and $\xi_1 = \psi^{-1}(x_1), \xi_N = \psi^{-1}(x_N)$. Thus, the interval of input points changes to $[\xi_1, \xi_N]$ from $[x_1, x_N]$. However, if we perform such a transformation for every sub-layer, the scale of the input becomes larger and inconsistent, making the network converge slower Ioffe (2015). Herein, we argue that the normalization for the input of every layer is necessary, and Bozorgasl & Chen (2024) already utilizes the batch normalization Ioffe (2015) on every layer to enhance the performance of KANs. In our SincKAN, a normalizing transformation, $\phi(x) = \frac{x-\mu}{\sigma}$ is introduced, where $\sigma$ is the scaling factor and $\mu$ is the shifting factor. Composing $\phi$ and $\psi$ still meets the condition of $\psi$ in Theorem 2 and the transformed function $f(\psi \circ \phi^{-1}(\xi))$ also satisfies assumptions 1 and 2 in Theorem 1 with $\alpha^\star, \beta^\star$ and $d^\star$ (proved in Appendix A). Consequently, the optimal value of the step size $h$ is changed to $\sqrt{\pi d^\star / \beta^\star N}$.

Let us define the normalized coordinate transformation $\gamma^{-1}(x) := \phi \circ \psi^{-1}(x)$ such that $\gamma^{-1} : (a, b) \to (-\infty, \infty)$ and $[x_1, x_N] \to \left[\frac{\xi_1 - \mu}{\sigma}, \frac{\xi_N - \mu}{\sigma}\right]$, where $\sigma, \mu$ satisfies $\left[\frac{\xi_1 - \mu}{\sigma}, \frac{\xi_N - \mu}{\sigma}\right] \subset [-1, 1]$. Herein, instead of coordinate transformation $\psi$, we use normalized coordinate transformation $\gamma$ in SincKAN. As for the changing of optimal $h$ with different $\sigma, \mu$, the summation of different $h_j$ implemented in SincKAN makes it easy to meet the fixed scale $[-1, 1]$ *i.e.* we can depend the set $\{h_j\}$ on the domain $[-1, 1]$ regardless of $\sigma, \mu$.

**Exponential decay**   In Theorem 2, $f$ should satisfy the condition of exponential decay which constrains that $f(-\infty) = f(+\infty) = 0$. To utilize the Sinc methods on general functions, Richardson & Trefethen (2011) interpolates the subtraction $g - f$ instead of $f$, where $g$ is the linear function that has the same value as $f$ at the endpoints. In our SincKAN, we introduce a learnable linear function as a skip-connection to approximate the subtraction.

Finally, combining the three aforementioned approaches, we can define our learnable activation function in SincKAN:

$$\phi_{\text{sinc}}(x) = c_1 x + c_2 + \sum_{j=1}^{M} \sum_{i=-N}^{N} c_{i,j} S(i, h_j)(\gamma^{-1}(x)), \tag{14}$$

where $c_1, c_2, c_{i,j}$ are the learnable parameters $S$ is the Sinc function and $\gamma$ is the normalized transformation.

## 3 EXPERIMENTS

In this section, we will demonstrate the performance of SincKANs through experiments including approximating functions and solving PDEs, compared with several other representative networks: Multilayer perceptron (MLP) which is the classical and most common network used in PINNs, Modified MLP which is proposed to project the inputs to a high-dimensional feature space to enhance the hidden layers' capability, KAN which is proposed to replace MLP in AI for Science, and ChebyKAN which is proposed to improve the performance by combining KAN with the known approximation capabilities of Chebyshev polynomials and has already been examined in Shukla et al. (2024). In this paper, we choose to implement the normalized transformation $\gamma(x) = \tanh(x)$, and the linear skip connection $w_1 \in \mathbb{R}^{n_{in} \times n_{out}}, w_2 \in \mathbb{R}^{n_{out}}$. Note that, we also observed the instability of ChebyKAN highlighted in Shukla et al. (2024), and the ChebyKAN used in our experiments is actually the modified ChebyKAN proposed by Shukla et al. (2024) which has $\tanh$ activation function between each layers. The rest details of the used networks are provided in Appendix G. The other details including hyperparameters can be found in Appendix D.

### 3.1 LEARNING FOR APPROXIMATION

Approximating a function by given data is the main objective of KANs with applications in identifying relevant features, revealing modular structures, and discovering symbolic formulas Liu et al. (2024a). Additionally, in deep learning, the training process of a network can be regarded as approximating the map between complex functional spaces, thus the accuracy of approximation directly indicates the capability of a network. Therefore, we start with experiments on approximation to show the capability of SincKAN and verify whether SincKAN is a competitive network. In this section, to have consistent results with KAN, we inherit the metric RMSE which is used in KAN.

Sinc numerical methods are recognized theoretically and empirically as a powerful tool when dealing with singularities. However, in machine learning instead of numerical methods, we argue that SincKAN can be implemented in general cases. To demonstrate that SincKAN is robust, we conducted a series of experiments on both smooth functions which cubic splines interpolation is good at and singularity functions which Sinc interpolation is good at. We demonstrate partial of them in Table 1, the rest can be found in Table 5. The details of the used functions can be found in Appendix B.

Table 1: RMSE of functions for approximation

| **Function name** | **MLP** | **modified MLP** | **KAN** | **ChebyKAN** | **SincKAN (ours)** |
|---|---|---|---|---|---|
| *sin-low* | $1.51e\text{-}2 \pm 2.01e\text{-}2$ | $7.29e\text{-}4 \pm 2.98e\text{-}4$ | $1.27e\text{-}3 \pm 3.13e\text{-}4$ | $1.76e\text{-}3 \pm 3.19e\text{-}4$ | $3.55e\text{-}4 \pm 3.08e\text{-}4$ |
| *sin-high* | $7.07e\text{-}1 \pm 6.44e\text{-}8$ | $7.07e\text{-}1 \pm 1.15e\text{-}5$ | $7.06e\text{-}1 \pm 1.36e\text{-}3$ | $5.70e\text{-}2 \pm 5.99e\text{-}3$ | $3.94e\text{-}2 \pm 5.36e\text{-}3$ |
| *multi-sqrt* | $2.06e\text{-}3 \pm 1.16e\text{-}3$ | $4.59e\text{-}4 \pm 4.86e\text{-}4$ | $3.61e\text{-}4 \pm 8.67e\text{-}5$ | $2.34e\text{-}3 \pm 1.17e\text{-}3$ | $2.14e\text{-}4 \pm 2.49e\text{-}4$ |
| *piece-wise* | $2.01e\text{-}2 \pm 5.16e\text{-}3$ | $3.76e\text{-}2 \pm 1.83e\text{-}2$ | $5.84e\text{-}2 \pm 1.03e\text{-}2$ | $7.28e\text{-}3 \pm 9.59e\text{-}4$ | $2.14e\text{-}3 \pm 7.76e\text{-}4$ |
| *spectral-bias* | $4.18e\text{-}3 \pm 1.18e\text{-}3$ | $1.59e\text{-}3 \pm 2.39e\text{-}4$ | $4.73e\text{-}2 \pm 9.94e\text{-}3$ | $5.60e\text{-}3 \pm 2.56e\text{-}4$ | $1.48e\text{-}3 \pm 1.82e\text{-}4$ |
| *multimodal2-2d* | $7.55e\text{-}3 \pm 2.59e\text{-}3$ | $2.43e\text{-}3 \pm 1.15e\text{-}3$ | $7.97e\text{-}3 \pm 4.82e\text{-}3$ | $6.12e\text{-}2 \pm 2.02e\text{-}2$ | $2.11e\text{-}3 \pm 3.57e\text{-}4$ |
| *fractal-2d* | $2.89e\text{-}2 \pm 2.40e\text{-}2$ | $2.60e\text{-}2 \pm 7.26e\text{-}3$ | $2.54e\text{-}1 \pm 1.88e\text{-}2$ | $6.14e\text{-}2 \pm 5.07e\text{-}3$ | $7.53e\text{-}3 \pm 6.09e\text{-}4$ |
| *lpmv* | $1.27e\text{-}3 \pm 6.13e\text{-}5$ | $3.79e\text{-}4 \pm 1.58e\text{-}4$ | $4.19e\text{-}4 \pm 1.14e\text{-}4$ | $8.42e\text{-}3 \pm 1.29e\text{-}3$ | $2.72e\text{-}4 \pm 1.97e\text{-}5$ |
| *ellipj* | $6.51e\text{-}4 \pm 2.40e\text{-}5$ | $7.75e\text{-}5 \pm 9.74e\text{-}7$ | $1.29e\text{-}4 \pm 2.00e\text{-}5$ | $4.80e\text{-}3 \pm 4.03e\text{-}4$ | $3.02e\text{-}5 \pm 5.08e\text{-}6$ |
| *exp-sin-4d* | $3.85e\text{-}2 \pm 5.27e\text{-}4$ | $1.53e\text{-}2 \pm 1.33e\text{-}3$ | $6.22e\text{-}3 \pm 1.23e\text{-}3$ | $4.96e\text{-}1 \pm 1.91e\text{-}2$ | $2.74e\text{-}3 \pm 3.29e\text{-}4$ |
| *exp-100d* | $3.99e\text{-}3 \pm 1.88e\text{-}4$ | $1.04e\text{-}1 \pm 4.96e\text{-}4$ | $9.42e\text{-}2 \pm 3.02e\text{-}3$ | *nan* | $2.91e\text{-}3 \pm 1.03e\text{-}5$ |

The results in Table 1 show that SincKAN achieves impressive performance on low-frequency functions (*sin-low*), high-frequency functions (*sin-high*), continuous but non-differentiable functions (*multi-sqrt*) and discontinuous functions (*piece-wise*). Furthermore, the last function (*spectral-bias*) is designed to evaluate the ability to address the prevalent phenomenon of spectral bias by Rahaman et al. (2019), and the corresponding result in Table 1 indicates that SincKAN maximally alleviates the spectral bias. Additionally, we also evaluate every network on the finer grid to test their general-

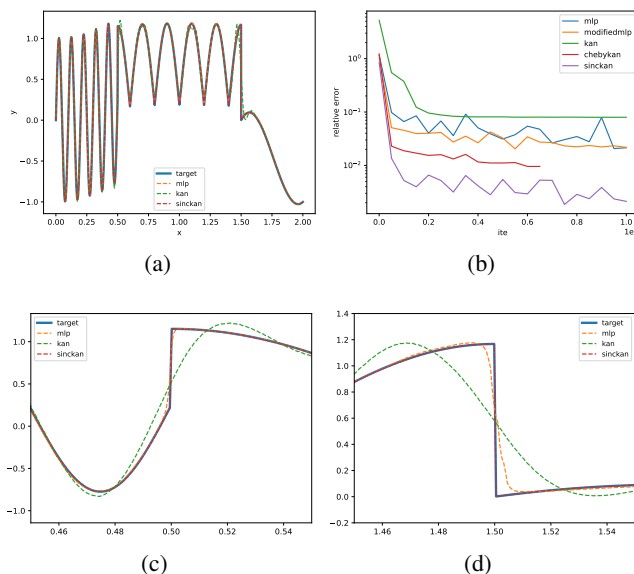

Figure 2: Fig. 2(a) depicts the function of *piece-wise* in Table 1 and compares the performance of SincKAN with MLP and KAN. Fig. 2(b) demonstrates the convergence of relative error for all networks, note that although the ChebyKAN we used is the modified ChebyKAN, its training is still unstable. Herein, the results of ChebyKAN in this paper are always the last valid error. Fig. 2(c) and Fig. 2(d) demonstrate the singularities in detail and show that the SincKAN can approximate the singularities well while MLP and KAN have obvious differences.

ization, we put the results on Appendix F. The comparison of cost is in Appendix H. Furthermore, as an important aspect of understanding SincKANs, we plot some interior $\phi$ in Appendix E.

### 3.1.1 SELECTING H

Utilizing a set of $\{h_i\}$ instead of a single step size $h$ is a novel approach that we developed specifically for SincKANs. To evaluate the effectiveness of this approach, in this section, we design a comprehensive experiment. Suppose $h_{min} = \min\{h_i\}, h_{max} = \max\{h_i\}$, based on the discussion of Section 2.3, the ideal case is the optimal $h^* = \sqrt{\pi d^\star/\beta^\star N} \in (h_{min}, h_{max})$. In the experiments, we provide two types of the set with two hyperparameters: the base number $h_0$ and the cardinality of the set $M$:

1. inverse decay $\{h_i\}_{i=1}^{M}$: $h_i = 1/ih_0$,
2. exponential decay $\{h_i\}_{i=1}^{M}$: $h_i = 1/h_0^i$.

We train SincKAN on *sin-low* and *sin-high* functions with $M = 1, 6, 12, 24$ for inverse decay and $M = 1, 2, 3$ for exponential decay and $h_0 = 2.0, \pi, 6.0, 10.0$. Besides, the number of discretized points $N_{points}$ and the degree $N_{degree}$ ($N_{degree} = 2N + 1$, where $N$ is the notation in Eq. (14)) also influence the performance of SincKAN for different $\{h_i\}_{i=1}^{M}$, we empirically set $N_{degree} = 100$, and $N_{points} = 5000$ in this experiment.

The results are illustrated in Fig. 12 for inverse decay and Fig. 13 for exponential decay, and the details including the corresponding error bars are shown in Appendix K. For sin-low function, the best RMSE $1.49e\text{-}4 \pm 8.74e\text{-}5$ is observed with $h_0 = 10.0$ and $M = 1$; for sin-high function, the best RMSE $4.60e\text{-}3 \pm 3.70e\text{-}4$ is observed in inverse decay with $h_0 = 10.0$ and $M = 24$. The experiments use two divergent Fourier spectra ($4\pi$, and $400\pi$), and get extremely different optimal hyperparameters $M$. But the RMSE is accurate enough for both sin-low and sin-high in inverse decay with $M = 6$ and $h_0 = 10.0$.

We also discuss the relationship between degree and data size in Appendix I and the update of basis in Appendix J.

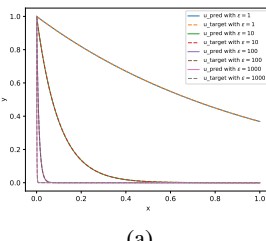 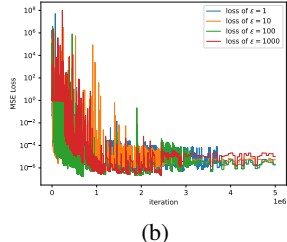 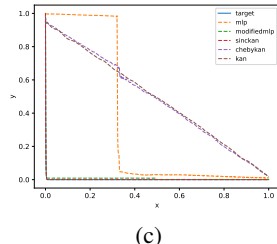

(a) (b) (c)

Figure 3: Fig. 3(a) depicts the exact solution of Eq. (15) with different $\epsilon$ and the corresponding predicted solution by SincKAN, states that SincKAN can solve Eq. (15) properly even with an extremely narrow boundary layer. Fig. 3(b) depicts the convergence of the training loss function of SincKAN with different $\epsilon$. Fig. 3(c) demonstrates the poor performance of different networks when solving Eq. (15) with $\epsilon = 1000$ due to the large derivatives of the boundary layer.

## 3.2 LEARNING FOR PIKANs

Solving PDEs is the main part of scientific computing, and PINNs are the representative framework for solving PDEs by neural networks. In this section, we solve a series of challenging PDEs to showcase the performance of SincKAN. At first, we select several classical PDEs to verify the robustness of SincKAN, the results are shown in Table 2, and the details of the PDEs can be found in Appendix C.

Table 2: Relative L2 error for chosen PDE problems

| Experiments | MLP | modified MLP | KAN | ChebyKAN | SincKAN (ours) |
|---|---|---|---|---|---|
| *perturbed* | $2.89e\text{-}2 \pm 3.09e\text{-}2$ | $6.30e\text{-}1 \pm 1.14e\text{-}1$ | $4.48e\text{-}3 \pm 4.20e\text{-}3$ | $6.73e\text{-}1 \pm 1.02e\text{-}1$ | $1.88e\text{-}3 \pm 8.55e\text{-}4$ |
| *nonlinear* | $3.92e\text{-}1 \pm 2.36e\text{-}5$ | $1.56e\text{-}2 \pm 2.10e\text{-}2$ | $6.15e\text{-}4 \pm 7.96e\text{-}4$ | $7.78e\text{-}1 \pm 2.67e\text{-}2$ | $1.77e\text{-}3 \pm 1.06e\text{-}3$ |
| *bl-2d* | $2.38e\text{-}1 \pm 6.22e\text{-}2$ | $5.34e\text{-}2 \pm 1.91e\text{-}2$ | $1.19e\text{-}2 \pm 4.22e\text{-}3$ | $5.97e\text{-}2 \pm 3.83e\text{-}2$ | $2.31e\text{-}3 \pm 7.10e\text{-}4$ |
| *ns-tg-u* | $8.14e\text{-}5 \pm 1.96e\text{-}6$ | $2.14e\text{-}5 \pm 2.67e\text{-}6$ | $3.21e\text{-}4 \pm 2.02e\text{-}5$ | $6.43e\text{-}2 \pm 2.70e\text{-}2$ | $6.51e\text{-}4 \pm 7.03e\text{-}5$ |
| *ns-tg-v* | $8.30e\text{-}5 \pm 2.47e\text{-}6$ | $1.91e\text{-}5 \pm 1.63e\text{-}6$ | $4.04e\text{-}4 \pm 1.25e\text{-}4$ | $5.86e\text{-}2 \pm 4.15e\text{-}2$ | $1.34e\text{-}3 \pm 4.38e\text{-}4$ |

### 3.2.1 BOUNDARY LAYER PROBLEMS

To intuitively show the performance of SincKAN compared with other networks, we conducted additional experiments on the boundary layer problem:

$$u_{xx}/\epsilon + u_x = 0, x \in [0, 1] \tag{15}$$

with the exact solution $u(x) = \exp(-\epsilon x)$. As $\epsilon$ increases, the width of the boundary layer (left) decreases, and the complexity of learning increases. The results shown in Table 3 and Fig. 4 reveal that SincKANs can handle the boundary layer effectively, while other networks struggle when $\epsilon$ is large.

Table 3: Relative L2 error for different $\epsilon$ in Eq. (15)

| $\epsilon$ | MLP | Modified MLP | KAN | ChebyKAN | SincKAN (ours) |
|---|---|---|---|---|---|
| 1 | $6.60e\text{-}5 \pm 1.91e\text{-}5$ | $3.88e\text{-}6 \pm 7.22e\text{-}7$ | $5.97e\text{-}6 \pm 5.24e\text{-}6$ | $1.98e\text{-}6 \pm 4.51e\text{-}7$ | $7.78e\text{-}5 \pm 1.14e\text{-}4$ |
| 10 | $2.83e\text{-}4 \pm 4.22e\text{-}5$ | $1.69e\text{-}4 \pm 6.85e\text{-}5$ | $3.23e\text{-}5 \pm 1.81e\text{-}5$ | $4.45e\text{-}6 \pm 4.01e\text{-}7$ | $1.14e\text{-}4 \pm 1.64e\text{-}4$ |
| $10^2$ | $1.29e\text{-}3 \pm 3.24e\text{-}4$ | $6.25e\text{-}4 \pm 2.27e\text{-}4$ | $1.25e\text{-}2 \pm 2.62e\text{-}3$ | $5.27e\text{-}4 \pm 6.55e\text{-}4$ | $1.68e\text{-}4 \pm 6.16e\text{-}5$ |
| $10^3$ | $9.87 \pm 8.70$ | $1.53e\text{-}1 \pm 5.59e\text{-}2$ | $11.3 \pm 8.79$ | $10.9 \pm 7.18$ | $5.48e\text{-}3 \pm 3.45e\text{-}3$ |
| $10^4$ | $8.41 \pm 3.64$ | $6.52 \pm 4.35$ | $9.73 \pm 8.77$ | $22.1 \pm 4.49$ | $5.27e\text{-}3 \pm 1.29e\text{-}3$ |

### 3.2.2 FRACTIONAL PDEs

Fractional PDEs are a challenge problem which the traditional numerical methods still face challenges due to their non-locality and singularity until now. We consider the following fractional PDE and demonstrate the results Fig. 4. The details are in Appendix D.3.

$$-\Delta^s u + \gamma u = f, \quad x \in (-1, 1), \tag{16}$$

with $f = 1 + \gamma u$, and

$$u(x) = \frac{2^{-2s}\Gamma(d/2)}{\Gamma(d/2+s)\Gamma(1+s)}(1 - \|x\|_2^2)^s, \tag{17}$$

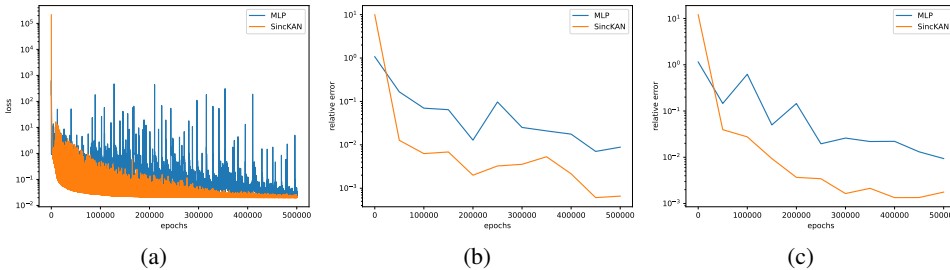

(a)          (b)          (c)

Figure 4: Fig. 4(a) depicts the loss of MLP and SincKAN with $s = 0.85$. For comparing the error between MLP and SincKAN, Fig. 4(b) depicts the results with $s = 0.85$, Fig. 4(c) depicts the results with $s = 0.95$.

### 3.2.3 HIGH-DIMENSIONAL PROBLEMS

We consider the classical high-dimensional Poisson equations:

$$\Delta u = f, \quad \boldsymbol{x} \in [0, 1]^d, \tag{18}$$

If $f = 2\alpha(2\alpha\|\boldsymbol{x}\|_2^2 - d)e^{-\alpha\|\boldsymbol{x}\|_2^2}$, the exact solution is $u(\boldsymbol{x}) = e^{-\alpha\|\boldsymbol{x}\|_2^2}$. In our experiments, we set $d = 100$. The details can be found in Appendix D, and Fig. 5 shows the performance of MLP and SincKAN.

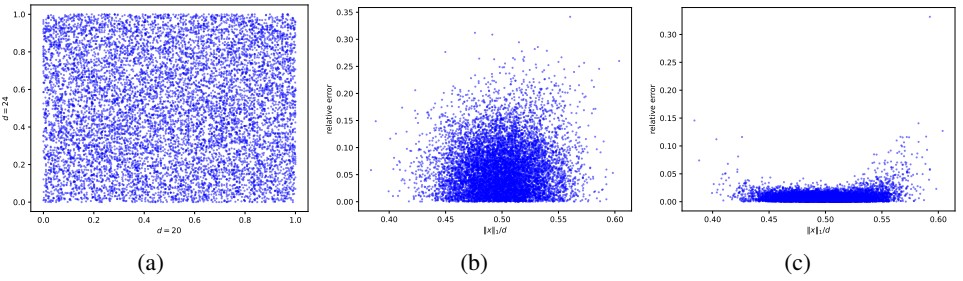

(a)          (b)          (c)

Figure 5: Fig. 5(a) shows the distribution of our testing points in the 20-th and 24-th dimension. Fig. 5(b) and Fig. 5(c) show the relative error of every points in MLP and SincKAN.

### 3.2.4 ABLATION STUDY

Compared with Sinc numerical methods, SincKANs have a normalized transformation; compared to KANs, SincKANs have a skip connection with linear functions. However, the Sinc numerical methods also have some choices of coordinate transformations and KANs also have a skip connection with SiLU functions. Herein, we conduct an ablation study on SincKANs with non-normalized

Table 4: Relative L2 error for ablation study

| $\psi$ | $\gamma$ | Linear | SiLU | T-nonlinear | Burgers' equation |
|---|---|---|---|---|---|
| ✗ | ✗ | ✗ | ✗ | $9.20e\text{-}4 \pm 4.31e\text{-}4$ | $1.57e\text{-}2 \pm 4.31e\text{-}3$ |
| ✗ | ✔ | ✗ | ✗ | $5.80e\text{-}4 \pm 1.89e\text{-}4$ | $6.21e\text{-}4 \pm 1.96e\text{-}4$ |
| ✗ | ✔ | ✔ | ✗ | $2.44e\text{-}4 \pm 6.08e\text{-}5$ | $3.12e\text{-}3 \pm 2.48e\text{-}3$ |
| ✗ | ✔ | ✗ | ✔ | $1.60e\text{-}4 \pm 3.17e\text{-}5$ | $8.90e\text{-}3 \pm 6.76e\text{-}3$ |
| ✔ | ✗ | ✔ | ✗ | $1.11e\text{-}2 \pm 1.17e\text{-}3$ | $7.36e\text{-}2 \pm 2.02e\text{-}2$ |
| ✔ | ✗ | ✗ | ✔ | $1.30e\text{-}2 \pm 4.00e\text{-}4$ | $1.12e\text{-}1 \pm 6.80e\text{-}3$ |

transformation and the SiLU skip connection to verify the effect of the two proposed modules. This experiment uses Burger's equation Eq. (46) and time-dependent nonlinear equation Eq. (48).

Table 4 shows the results of the ablation study where $\psi(x) = \log(\frac{x-a}{b-x})$, $\gamma(x) = \tanh(x)$, $\text{Linear}(x) = w_1 x + w_2 + \phi(x)$, and $\text{SiLU}(x) = w_b \text{silu}(x) + w_s \phi(x)$. The non-normalized transformation performs poorly, even compared to the cases without transformations. Although the linear skip connection is not the best for both equations, it is the most stable approach for SincKANs.

## 4   CONCLUSION

In this paper, we propose a novel network called Sinc Kolmogorov-Arnold Networks (SincKANs). Inspired by KANs, SincKANs leverage the Sinc functions to interpolate the activation function and successfully inherit the capability of handling singularities. To set the optimal $h$, we propose the multi-$h$ interpolation, and the corresponding experiments indicate that this novel approach is the main reason for SincKANs' superior ability in approximating complex smooth functions; to choose a proper coordinate transformation for machine learning, we propose the normalized transformation which prevents slow convergence.; to satisfy the decay condition, we introduce the skip-connection with learnable linear functions. After tackling the aforementioned challenges, SincKANs become a competitive network that can replace the current networks used for PINNs.

We begin with training on approximation problems to demonstrate the capability of SincKANs. The results reveal that SincKANs excel in most experiments compared with other networks. However, directly approximating the target function is an impractical objective for almost all machine learning tasks. After verifying the capability, we turn to solving PDEs in the PINNs framework. Although the SincKANs achieve impressive performance in approximation tasks for solving all chosen PDEs, SincKANs merely have the best accuracy on boundary layer problems, due to the oscillations caused by the inaccuracy of derivatives.

**Limitations**: Approximating derivative by Sinc numerical methods is always inaccurate in the neighborhood of the Sinc end-points. To address this problem, Stenger (2009) suggested using Lagrange polynomial to approximate the derivative instead of straightforwardly calculating the derivative of Sinc polynomials, Wu et al. (2006) used several discrete functions to replace the derivative of Sinc polynomials, etc. Unfortunately, to the best of our knowledge, there isn't an approach that can be implemented in our SincKAN when we demand the derivatives of SincKAN in PIKANs. Herein, to alleviate the inaccuracy, we choose small $h_0$, small $M$, and small $N$ so that SincKAN can solve PDEs, otherwise, the solution will have oscillations (see Appendix L). Such kind of setting limits the capability of SincKAN and we argue that this is the main reason that SincKAN can obtain good results but not the best results for some cases. Furthermore, the inaccuracy limits SincKAN in solving high-order problems such as Korteweg–De Vries equations, and Kuramoto–Sivashinsky equations.

**Futures**: However, as the accuracy of approximating the derivative decreases with the order of derivative increases if the PDE merely requires the first derivatives, then the SincKANs will release the limitation to have larger enough $h_0$, $M$, and $N$ and improve the performance. In literature, to avoid calculating the high-order derivatives, MIM Lyu et al. (2022); Li et al. (2024) is proposed to use the mixed residual method which transforms a high-order PDE into a first-order PDE system. SincKANs can implement this approach to calculate several first-order derivatives instead of the high-order derivatives so that SincKANs can have accurate estimations for the residual loss. Furthermore, replacing the automatic differentiation Cen & Zou (2024); Yu et al. (2024) by other operators is also expected.

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

## A  PROOF OF TRANSFORMATION

**Theorem 3.** *For a linear transformation $x = \sigma\xi + \mu$, if $f(x), \quad x \in \mathbb{R}$, satisfies the assumption of Theorem 1 for some $d, \alpha, \beta > 0$, that is,*

*(1) $f$ belongs to $H^1(\mathcal{D}_d)$, where $H^1$ is the Hardy space and $\mathcal{D}_d = \{z \in \mathbb{C} \mid |\Im z| < d\}$;*

*(2) $f$ decays exponentially on the real line, that is, $|f(x)| \le \alpha \exp(-\beta|x|), \forall x \in \mathbb{R}$.*

*the transformed function $\tilde{f}(\xi) = f(\sigma\xi + \mu), \xi \in \mathbb{R}$ satisfies assumptions 1 and 2 with some $\tilde{\alpha}, \tilde{\beta}$ and $\tilde{d}$.*

*Proof.* For assumption (1), as $\mu$ and $\sigma$ are real number,

$$f(x) \in H^1(\mathcal{D}_d) \tag{19}$$

$\Leftrightarrow$

$$\lim_{\varepsilon \to 0} \int_{\partial\mathcal{D}_d(\varepsilon)} |f(z)||\mathrm{d}z| < \infty \tag{20}$$

$\Leftrightarrow$

$$\int_{-\infty}^{\infty} |f(x \pm id)|\mathrm{d}x < \infty \tag{21}$$

Then

$$\int_{-\infty}^{\infty} |\tilde{f}(\tilde{x} \pm i\tilde{d})|\mathrm{d}\tilde{x}$$

$$= \int_{-\infty}^{\infty} |f(\sigma\tilde{x} + \mu \pm i\sigma\tilde{d})|\mathrm{d}x \tag{22}$$

$$= \int_{-\infty}^{\infty} |f(x' \pm id')|\mathrm{d}x', \text{ where } x' = \sigma\tilde{x} + \mu, d' = \sigma\tilde{d}$$

If $d' < d$, i.e. $\tilde{d} < \frac{d}{\sigma}$, then $\int_{-\infty}^{\infty} |f(x' \pm id')|\mathrm{d}x' < \infty \Rightarrow \tilde{f}(\xi) \in H^1(\mathcal{D}_{\tilde{d}})$. As $\frac{d}{\sigma} > 0$, $\tilde{d}$ exists, (1) is satisfied.

For assumption (2), as $|\tilde{f}(\xi)| \le \alpha\exp(-\beta|\sigma\xi + \mu|), \forall \xi \in \mathbb{R}$, if there exists $\tilde{\alpha}, \tilde{\beta} > 0$ such that $\alpha\exp(-\beta|\sigma\xi + \mu|) \le \tilde{\alpha}\exp(-\tilde{\beta}|\xi|)$, then (2) is satisfied:

$$\alpha\exp(-\beta|\sigma\xi + \mu|) \le \tilde{\alpha}\exp(-\tilde{\beta}|\xi|) \tag{23}$$

$\Rightarrow$

$$\log\alpha - \beta|\sigma\xi + \mu| \le \log\tilde{\alpha} - \tilde{\beta}|\xi| \tag{24}$$

$\Rightarrow$

$$\tilde{\beta} \le \frac{\log\frac{\tilde{\alpha}}{\alpha}}{|\xi|} + \beta\left|\sigma + \frac{\mu}{\xi}\right|, \text{ if } \xi \ne 0. \tag{25}$$

Thus, if $\tilde{\alpha} > \alpha$, there exists a $\tilde{\beta} > 0$ that satisfies the above inequality; on the other hand if $\xi = 0$, obviously $\alpha\exp(-\beta|\sigma\xi + \mu|) \le \alpha < \tilde{\alpha}$. Herein, $\tilde{\alpha}, \tilde{\beta}$ exists.

In total, there exists $\tilde{d}, \tilde{\alpha}, \tilde{\beta} > 0$ such that $\tilde{f}$ satisfies assumption (1) and (2) $\qquad\square$

## B    EXPLICIT EXPRESSION OF FUNCTIONS

The following functions are used in Table 1 and Table 5.

1. *sin-low*
$$f(x) = \sin(4\pi x), \quad x \in [-1, 1] \tag{26}$$

2. *sin-high*
$$f(x) = \sin(400\pi x), \quad x \in [-1, 1] \tag{27}$$

3. *bl*
$$f(x) = e^{-100x}, \quad x \in [0, 1] \tag{28}$$

4. *sqrt*
$$f(x) = \sqrt{x}, \quad x \in [0, 1] \tag{29}$$

5. *double-exponential*
$$f(x) = \frac{x(1-x)\mathrm{e}^{-x}}{(1/2)^2 + (x - 1/2)^2}, \quad x \in [0, 1] \tag{30}$$

6. *multi-sqrt*
$$f(x) = x^{1/2}(1-x)^{3/4}, \quad x \in [0, 1] \tag{31}$$

7. *piece-wise*
$$f(x) = \begin{cases} \sin(20\pi x) + x^2, & x \in [0, 0.5] \\ 0.5xe^{-x} + |\sin(5\pi x)|, & x \in [0.5, 1.5] \\ \log(x-1)/\log(2) - \cos(2\pi x), & x \in [1.5, 2] \end{cases} \tag{32}$$

8. *spectral-bias*
$$f(x) = \begin{cases} \sum_{k=1}^{4} \sin(kx) + 5, & x \in [-1, 0] \\ \cos(10x), & x \in [0, 1] \end{cases} \tag{33}$$

9. *multimodal1-2d*
$$f(\boldsymbol{x}) = -|\sin(x_1) \cdot \cos(x_2)| \cdot \exp\left(\left|1 - \frac{\sqrt{x_1^2 + x_2^2}}{\pi}\right|\right), \quad \boldsymbol{x} \in [0, 1]^2 \tag{34}$$

10. *multimodal2-2d*
$$\begin{aligned} f(\boldsymbol{x}) = &-20e^{-0.2\sqrt{(x_1^2 + x_2^2)}} - e^{\cos(2\pi x_1) + \cos(2\pi x_2)} \\ &+ e + 20, \quad \boldsymbol{x} \in [-1, 1]^2 \end{aligned} \tag{35}$$

11. *fractal-2d*
$$\begin{aligned} f(\boldsymbol{x}) = &\left[\sin(10\pi x_1)\cos(10\pi x_2) + \sin\left(\pi\left(x_1^2 + x_2^2\right)\right) + |x_1 - x_2| + \frac{\sin(5x_1 x_2)}{0.1 + |x_1 + x_2|}\right] \\ &\exp\left(-0.1\left(x_1^2 + x_2^2\right)\right), \quad \boldsymbol{x} \in [0, 1]^2 \end{aligned} \tag{36}$$

12. *lpmv*
$$f(\boldsymbol{x}) = \text{scipy.special.lpmv}(1, x_1, x_2), \quad \boldsymbol{x} \in [0, 1]^2 \tag{37}$$

13. *ellipj*
$$f(\boldsymbol{x}) = \text{scipy.special.ellipj}(x_1, x_2), \quad \boldsymbol{x} \in [0, 1]^2 \tag{38}$$

14. *sph-harm*
$$f(\boldsymbol{x}) = \text{scipy.special.sph\_harm}(1, 1, x_1, x_2), \quad \boldsymbol{x} \in [0, 1]^2 \tag{39}$$

15. *exp-sin-4d*
$$f(\boldsymbol{x}, \alpha) = \exp\left[0.5\sin\left(\pi \cdot \sum_{i=1}^{2} x_i^2\right) + 0.5\sin\left(\pi \cdot \sum_{i=3}^{4} x_i^2\right)\right], \quad \boldsymbol{x} \in [-1, 1]^4 \tag{40}$$

16. *exp-100d*
$$f(\boldsymbol{x}) = \exp\left(-0.001\|\boldsymbol{x}\|_2^2\right), \quad \boldsymbol{x} \in [0, 1]^{100} \tag{41}$$

Table 5: RMSE of functions for approximation

| Function name | MLP | modified MLP | KAN | ChebyKAN | SincKAN (ours) |
|---|---|---|---|---|---|
| *bl* | $7.59e\text{-}4 \pm 1.13e\text{-}3$ | $5.73e\text{-}4 \pm 4.06e\text{-}4$ | $2.54e\text{-}4 \pm 7.99e\text{-}5$ | $1.81e\text{-}3 \pm 6.98e\text{-}4$ | $4.76e\text{-}5 \pm 4.25e\text{-}5$ |
| *sqrt* | $3.06e\text{-}3 \pm 9.34e\text{-}4$ | $4.46e\text{-}5 \pm 5.51e\text{-}5$ | $4.79e\text{-}4 \pm 1.23e\text{-}4$ | $3.69e\text{-}3 \pm 1.27e\text{-}3$ | $3.24e\text{-}4 \pm 1.31e\text{-}4$ |
| *multimodal1-2d* | $2.69e\text{-}3 \pm 1.28e\text{-}3$ | $2.08e\text{-}4 \pm 5.48e\text{-}5$ | $1.23e\text{-}2 \pm 5.71e\text{-}4$ | $1.23e\text{-}2 \pm 5.83e\text{-}4$ | $2.36e\text{-}3 \pm 2.22e\text{-}3$ |
| *sph-harm* | $3.93e\text{-}4 \pm 2.27e\text{-}5$ | $4.90e\text{-}5 \pm 2.83e\text{-}5$ | $7.67e\text{-}5 \pm 6.05e\text{-}6$ | $5.13e\text{-}3 \pm 1.41e\text{-}5$ | $6.26e\text{-}5 \pm 1.40e\text{-}5$ |
| *double exponential* | $1.95e\text{-}3 \pm 8.17e\text{-}4$ | $7.77e\text{-}5 \pm 4.03e\text{-}5$ | $2.15e\text{-}4 \pm 1.52e\text{-}4$ | $3.11e\text{-}3 \pm 2.16e\text{-}3$ | $7.06e\text{-}5 \pm 1.09e\text{-}5$ |

## C    DETAILS OF PDES

### C.1    1D PROBLEMS

### C.2    PERTURBED BOUNDARY VALUE PROBLEM

We consider the singularly perturbed second-order boundary value problem (*perturbed* in Table 2):

$$\epsilon u_{xx} - u_x = f(x), \quad x \in [-1, 1]. \tag{42}$$

In specific cases, the problem has exact solutions, in this paper, we choose $f(x) = -1$, and the exact solution is

$$u(x) = 1 + x + \frac{e^{\frac{x}{\epsilon}} - 1}{e^{\frac{1}{\epsilon}} - 1}, \tag{43}$$

where $\epsilon = 0.01$ in our experiments.

### C.3    NONLINEAR PROBLEM

We consider the nonlinear boundary value problem (*nonlinear* in Table 2):

$$
\begin{aligned}
-u_{xx} + \frac{u_x}{x} + \frac{u}{x^2} &= \frac{\left(-41x^2 + 34x - 1\right)\sqrt{x}}{4} - 2x + \frac{1}{x^2}, \quad x \in [0, 1] \\
u(0) - 2u_x(0) &= 1, \\
3u(1) + u_x(1) &= 9,
\end{aligned}
\tag{44}
$$

with the exact solutions

$$u(x) = x^{5/2}(1 - x)^2 + x^3 + 1. \tag{45}$$

### C.4    BURGERS

We consider the Burgers' equation (**Burgers' equation** in Table 4):

$$\frac{\partial u}{\partial t} + u\frac{\partial u}{\partial x} - \nu\frac{\partial^2 u}{\partial x^2} = 0, \quad x \in [-1, 1], t \in [0, 0.1]. \tag{46}$$

with Dirichlet boundary condition, and the exact solution is

$$u = \frac{a}{2} - \frac{a \tanh\left(\frac{a(x - at/2)}{4\nu}\right)}{2}, \tag{47}$$

where $a = 0.5, \nu = 0.01$ in our experiments.

### C.5    T-NONLINEAR PROBLEM

We consider the time-dependent nonlinear problem (**T-nonlinear** in Table 4):

$$
\begin{aligned}
u_t &= \frac{x + 2}{t + 1}u_x, \quad x \in [-1, 1], t \in [0, 0.1]. \\
u(x, 0) &= \cos(x + 2), \\
u(1, t) &= \cos(3(t + 1)),
\end{aligned}
\tag{48}
$$

with the exact solution:

$$u(x, t) = \cos((t + 1)(x + 2)). \tag{49}$$

### C.6 CONVECTION-DIFFUSION

We consider the 1-D convection-diffusion equation with periodic boundary conditions (used in Appendix L):

$$u_t + au_x - \epsilon u_{xx} = 0, \quad x \in [-1, 1], t \in [0, 0.1],$$

$$u(x, 0) = \sum_{k=0}^{5} \sin(k\pi x), \tag{50}$$

with the analytic solution

$$u(x, t) = \sum_{k=0}^{5} \sin(k\pi x - ka\pi t) e^{-\epsilon k^2 \pi^2 t}, \tag{51}$$

where $\epsilon = 0.01$, and $a = 0.1$ in our experiments.

### C.7 2D PROBLEMS

#### C.7.1 BOUNDARY LAYER

We consider the 2-D boundary layer problem (*bl-2d* in Table 2):

$$u_{xx}/\alpha_1 + u_x + u_{yy}/\alpha_2 + u_y = 0, \tag{52}$$

with the exact solution

$$u(x, y) = \exp(-\alpha_1 x) + \exp(-\alpha_2 y), \tag{53}$$

where $\alpha_1 = \alpha_2 = 100$ in our experiments.

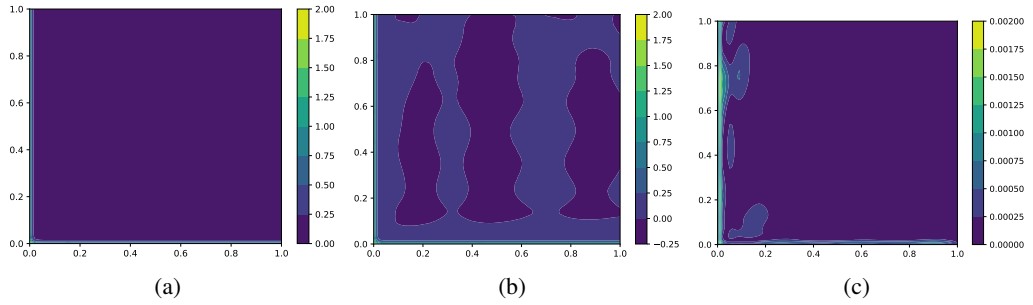

$$\text{(a)} \qquad\qquad \text{(b)} \qquad\qquad \text{(c)}$$

Figure 6: Fig. 6(a) depicts the exact solution of Eq. (52), Fig. 6(b) shows the solution predicted by SincKAN, Fig. 6(c) shows the absolute error between the predicted solution and the exact solution, exhibits that the error mainly comes from the boundary layer.

#### C.7.2 NAVIER STOKES EQUATIONS

We consider the Taylor–Green vortex (*ns-tg-u* and *ns-tg-v* in Table 2):

$$\nabla \cdot \boldsymbol{u} = 0, \quad t \in [0, T], \; \boldsymbol{x} \in \Omega,$$

$$\partial_t \boldsymbol{u} + \boldsymbol{u} \cdot \nabla \boldsymbol{u} = -\nabla p + \nu \triangle \boldsymbol{u}, \quad t \in [0, T], \; \boldsymbol{x} \in \Omega, \tag{54}$$

where $\boldsymbol{u} = (u, v)$, with the exact solution

$$u = -\cos(x)\sin(y)\exp(-2\nu t)$$

$$v = \sin(x)\cos(y)\exp(-2\nu t)$$

$$p = -(\cos(2x) + \sin(2y))\exp(-4\nu t)/4 \tag{55}$$

with $T = 1$, $\nu = 1/400$ in our experiments. After dimensionless, $\boldsymbol{x} \in [0, 1]^2$.

## D    EXPERIMENT DETAILS

Totally, in our experiments, the Adam Kingma & Ba (2014) optimizer is used with the exponential decay learning rate. The MLP and modified MLP are equipped with the tanh activations and Xavier initialization inherited from Raissi et al. (2019).

### D.1    APPROXIMATION

The hyperparameters of used networks are shown in Table 7.

- For the 1-D problem, we generate the training dataset by uniformly discretizing the input interval to 5000 points and train the network with 3000 points randomly sampled from the training dataset for each iteration. In total, We train every network with $10^5$ iterations. Additionally, to evaluate the generalization, we generate the testing (fine) dataset by uniformly discretizing the input interval to 10000 points.

### D.2    PIKANS

The hyperparameters of used networks are shown in Table 8.

- For time-independent 1-D problems, we generate the training dataset by uniformly discretizing the input interval to 1000 points, then train the network with 500 points randomly sampled from the training dataset for each iteration. In total, We train every network with $1.5 \times 10^6$ iterations.
- For time-dependent 1-D problems, we generate the training dataset by uniformly discretizing the spatial dimension to 1000 points and the temporal dimension to 11 points, then train the network with 5000 points randomly sampled from the training dataset for each iteration. In total, We train every network with $1.5 \times 10^6$ iterations.
- For time-independent 2-D problems, we generate the training dataset by uniformly discretizing every dimension to 100 points, then train the network with 5000 points randomly sampled from the training dataset for each iteration. In total, We train every network with $1.5 \times 10^6$ iterations.
- For time-dependent 2-D problems, we generate the training dataset by uniformly discretizing every spatial dimension to 100 points and the temporal dimension to 11 points, then train the network with 50000 points randomly sampled from the training dataset for each iteration. In total, We train every network with $1.5 \times 10^6$ iterations.
- For 100-D Poisson equation, we sample 2000 points from the interior domain, and 10000 points from the boundary domain for each iteration. We train every network with $5 \times 10^6$ iterations. For the testing data, we sample 8000 points from the interior domain and 2000 points from the boundary domain.

### D.3    FRACTIONAL PINNS

The framework is from fPINNs Pang et al. (2019), we utilize the second order Grünwald-Letnikov (GL) formula Lischke et al. (2020):

$$(-\Delta)^{\alpha/2} u(x_j) = (1-\beta)\delta^\alpha_{\Delta x, 1} u(x_j) + \beta \delta^\alpha_{\Delta x, 0} u(x_j) + O\left((\Delta x)^2\right), \tag{56}$$

where

$$\delta^\alpha_{\Delta x, p} u(x_j) := (\Delta x)^{-\alpha} \sum_{k=0}^{j} (-1)^k \binom{\alpha}{k} u(x_j - (k-p)\Delta x)$$
$$+ (\Delta x)^{-\alpha} \sum_{k=0}^{N-j} (-1)^k \binom{\alpha}{k} u(x_j + (k-p)\Delta x). \tag{57}$$

The interval is $(0,1)$ discretized by $\Delta x = 1/N$, where $N = 101$. The learning rate is $10^{-2}$ for SincKAN and $10^{-3}$ for MLP. The iteration number is $5 \times 10^5$. The shape of SincKAN is $8 \times 8 \times 1$. The performance is shown in Fig. 7.

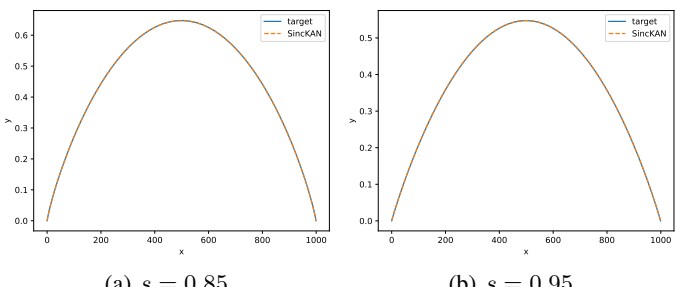

(a) $s = 0.85$ (b) $s = 0.95$

Figure 7: Fig. 7(a) and Fig. 7(b) show the target solution and the predicted solution by SincKAN.

## E INTERIOR APPROXIMATION OF SINCKAN

The results are demonstrated in Fig. 8.

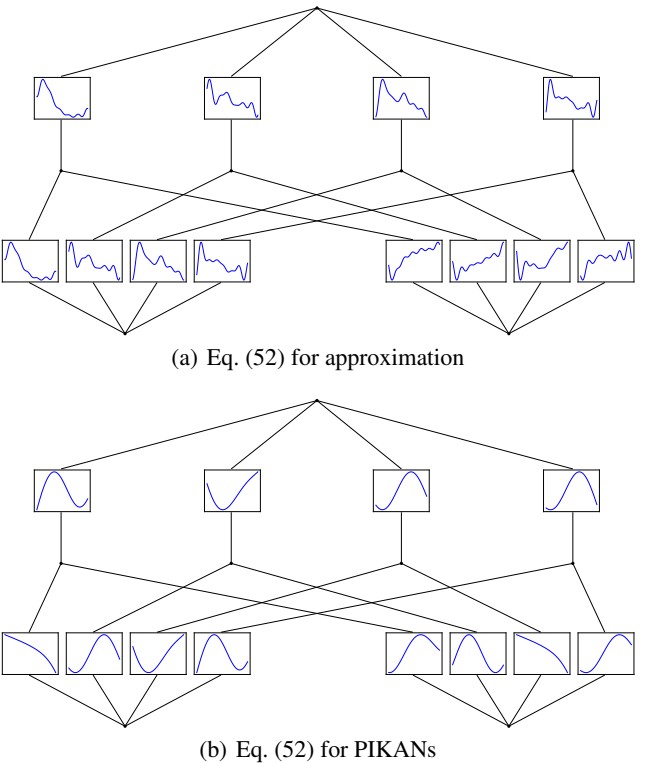

(a) Eq. (52) for approximation

(b) Eq. (52) for PIKANs

Figure 8: Fig. 8(a) used high $M$ so the interpolated $phi$ has oscillations while Fig. 8(b) uses low $M$ so the $phi$ is more smooth

## F APPROXIMATION ON FINE GRIDS

As we discussed in Appendix D, we additionally evaluate every network on fine grids. And due to the oscillations discussed in Appendix L, Table 6 reveals the weak generalization of SincKANs demands further research, although the applications of approximating a function don't strongly require this capability.

Table 6: RMSE evaluated on fine grids

| Function name | MLP | modified MLP | KAN | ChebyKAN | SincKAN (ours) |
|---|---|---|---|---|---|
| *sin-low* | $1.51e\text{-}2 \pm 2.01e\text{-}2$ | $7.29e\text{-}4 \pm 2.97e\text{-}4$ | $1.27e\text{-}3 \pm 3.04e\text{-}4$ | $1.76e\text{-}3 \pm 3.19e\text{-}4$ | $4.46e\text{-}4 \pm 2.79e\text{-}4$ |
| *sin-high* | $7.07e\text{-}1 \pm 4.21e\text{-}8$ | $7.07e\text{-}1 \pm 1.31e\text{-}5$ | $7.06e\text{-}1 \pm 1.29e\text{-}3$ | $5.70e\text{-}2 \pm 5.99e\text{-}3$ | $4.15e\text{-}2 \pm 4.53e\text{-}3$ |
| *bl* | $7.63e\text{-}4 \pm 1.13e\text{-}3$ | $5.72e\text{-}4 \pm 4.01e\text{-}4$ | $2.62e\text{-}4 \pm 7.89e\text{-}5$ | $1.81e\text{-}3 \pm 6.98e\text{-}4$ | $2.28e\text{-}4 \pm 1.16e\text{-}4$ |
| *double exponential* | $1.95e\text{-}3 \pm 8.17e\text{-}4$ | $7.76e\text{-}5 \pm 4.07e\text{-}5$ | $2.18e\text{-}4 \pm 1.51e\text{-}4$ | $3.11e\text{-}3 \pm 2.16e\text{-}3$ | $7.06e\text{-}5 \pm 1.09e\text{-}5$ |
| sqrt | $3.06e\text{-}3 \pm 9.34e\text{-}4$ | $4.46e\text{-}5 \pm 5.51e\text{-}5$ | $4.79e\text{-}4 \pm 1.23e\text{-}4$ | $3.69e\text{-}3 \pm 1.27e\text{-}3$ | $3.24e\text{-}4 \pm 1.31e\text{-}4$ |
| *multi-sqrt* | $2.06e\text{-}3 \pm 1.16e\text{-}3$ | $4.59e\text{-}4 \pm 4.86e\text{-}4$ | $3.61e\text{-}4 \pm 8.67e\text{-}5$ | $2.34e\text{-}3 \pm 1.17e\text{-}3$ | $2.14e\text{-}4 \pm 2.49e\text{-}4$ |
| *piece-wise* | $2.06e\text{-}2 \pm 5.57e\text{-}3$ | $3.75e\text{-}2 \pm 1.81e\text{-}2$ | $5.84e\text{-}2 \pm 1.03e\text{-}2$ | $7.28e\text{-}3 \pm 9.59e\text{-}4$ | $9.41e\text{-}3 \pm 2.14e\text{-}4$ |
| *spectral-bias* | $2.48e\text{-}2 \pm 9.77e\text{-}3$ | $1.88e\text{-}2 \pm 9.55e\text{-}4$ | $4.79e\text{-}2 \pm 9.44e\text{-}3$ | $2.18e\text{-}2 \pm 2.97e\text{-}4$ | $2.21e\text{-}2 \pm 9.98e\text{-}5$ |

## G  DETAILS OF OTHER NETWORKS

### G.1  MLP

Multilayer Perceptron (MLP) is the neural network consisting of fully connected neurons with a nonlinear activation function, and can be represented simply by:

$$\text{MLP}(x) = \left(W^{L-1} \circ \sigma \circ W^{L-2} \circ \sigma \circ \cdots \circ W^1 \circ \sigma \circ W^0\right) \boldsymbol{x} \tag{58}$$

where $W^i(x) = W_i x + b_i$, $W_i \in \mathbb{R}^{m_i \times n_i}$ is a learnable matrix, $b_i \in \mathbb{R}^{m_i}$ is a learnable bias, $\sigma$ is the chosen nonlinear activation function, and $L$ is the depth of MLP.

### G.2  MODIFIED MLP

Modified MLP is an upgraded network of MLP inspired by the transformer networks. It introduces two extra features and has a skip connection with them:

$$U = \sigma(W^{L+1}\boldsymbol{x}), \quad V = \sigma(W^{L+2}\boldsymbol{x}), \quad H^1 = \sigma(W^0\boldsymbol{x}),$$
$$H^{i+1} = (1 - \sigma(W^i H^i)) * U + \sigma(W^i H^i) * V, \quad i = 1, \cdots, L, \tag{59}$$
$$\text{ModifiedMLP}(\boldsymbol{x}) = W^{L+3} H^{L+1},$$

where $*$ is the element-wise multiplication.

### G.3  KAN

#### G.3.1  KOLMOGOROV-ARNOLD REPRESENTATION THEOREM

The Kolmogorov-Arnold representation theorem states that any multivariate continuous function on a bounded domain can be represented as a finite composition of univariate continuous functions and addition. Specifically for a continuous $f : [0,1]^n \to \mathbb{R}$, there exists continuous 1D functions $\phi_{q,p}$, $\Phi_q$ such that

$$f(\mathbf{x}) = f(x_1, \cdots, x_n) = \sum_{q=1}^{2n+1} \Phi_q \left( \sum_{p=1}^{n} \phi_{q,p}(x_p) \right). \tag{60}$$

#### G.3.2  KOLMOGOROV-ARNOLD NETWORK

Inspired by Kolmogorov-Arnold representation theorem, Kolmogorov-Arnold Network (KAN) is a novel network that aims to be more accurate and interpretable than MLP. The main difference is KAN's activation functions are learnable: suppose $\boldsymbol{\Phi} = \{\phi_{p,q}\}$ is the matrix of univariable functions where $p = 1, 2, \ldots n_{in}, q = 1, 2, \ldots n_{out}$ and $\theta$ represents the trainable parameters. The KAN can be defined by:

$$\text{KAN}(\boldsymbol{x}) = (\boldsymbol{\Phi}_{L-1} \circ \boldsymbol{\Phi}_{L-2} \circ \cdots \circ \boldsymbol{\Phi}_1 \circ \boldsymbol{\Phi}_0) \boldsymbol{x}, \quad \boldsymbol{x} \in \mathbb{R}^d, \tag{61}$$

where

$$\boldsymbol{\Phi}_l(\boldsymbol{x}^{(l)}) = \left\{ \sum_{i=1}^{n_{in}^{(l)}} \phi_{j,i}\left(x_i^{(l)}\right) \right\}_{j=1}^{n_{out}^{(l)}}, \quad \forall l = 0, 1, \cdots, L-1 \tag{62}$$

where $\boldsymbol{x}^l \in \mathbb{R}^{n_{in}^{(l)}}, \boldsymbol{\Phi}_l(\boldsymbol{x}^{(l)}) \in \mathbb{R}^{n_{out}^{(l)}}$. If a size of KAN is represented by an integer array $[n_0, n_1, \cdots, n_L]$, then $n_{in}^{(l)} = n_l, n_{out}^{(l)} = n_{l+1}$. To approximate every single activation function $\phi$, KAN utilizes the summation of basis function and spline interpolation:

$$\phi(x) = w_b \text{silu}(x) + w_s \left( \sum_i^N c_i B_i(x) \right), \tag{63}$$

where $c_i, w_s, w_b$ are learnable, $B_i$ is the $k$-th order B-splines, $N = G + k - 1$, and $G$ is the grid size.

## G.4 CHEBYKAN

ChebyKAN utilizes the Chebyshev polynomials to construct the learnable activation function $\phi$ in KAN. And the modified ChebyKAN embeds the tanh activation function between every layer. Thus the ChebyKAN used in our experiments can be defined by:

$$\text{ChebyKAN}(\boldsymbol{x}) = (\boldsymbol{\Phi}_{L-1} \circ \tanh \circ \boldsymbol{\Phi}_{L-2} \circ \cdots \circ \boldsymbol{\Phi}_1 \circ \tanh \circ \boldsymbol{\Phi}_0 \circ \tanh \circ) \boldsymbol{x}, \quad \boldsymbol{x} \in \mathbb{R}^d, \tag{64}$$

where $\boldsymbol{\Phi}$ has the same definition of Eq. (62) with different univariable function

$$\phi = \sum_i c_i T_i(x), \tag{65}$$

where $T_i$ is the $i$th Chebyshev polynomial.

# H COMPUTATIONAL COST

## H.1 TRAINING

In Eq. (14), SincKAN has an additional summation on several $h$, so the trainable coefficients $c$ are $M$ times larger than KAN and ChebyKAN. However, the training time is not only dependent on the number of total parameters, thus, we demonstrate the cost of training for approximation in Table 7, and demonstrate the cost of training for PDEs in Table 8. In Table 7 and Table 8, we use 'depth $\times$ width' to represent the size for MLP and modified MLP; 'width $\times$ degree' to represent the size for KAN and ChebyKAN; and 'width $\times$ degree $\times M$' to represent the size for SincKAN. Note that we train the network in two environments distinguished by two superscripts:

†: training on single NVIDIA A100-SXM4-80GB with CUDA version: 12.4.

‡: training on single NVIDIA A40-48GB with CUDA version: 12.4.

Table 7: Computational cost for approximation†

| Network | Size | Training rate (iter/sec) | Referencing time (ms) | Parameters |
|---|---|---|---|---|
| MLP | $10 \times 100$ | $9.89 \times 10^2$ | $1.62 \times 10^1$ | 81101 |
| modified MLP | $10 \times 100$ | $9.13 \times 10^2$ | $2.92 \times 10^1$ | 81501 |
| KAN | $8 \times 8$ | $1.15 \times 10^3$ | $1.01 \times 10^2$ | 160 |
| ChebyKAN | $40 \times 40$ | $1.29 \times 10^3$ | $3.11 \times 10^1$ | 3280 |
| SincKAN | $8 \times 100 \times 6$ | $1.29 \times 10^3$ | $2.06 \times 10^1$ | 9696 |

## H.2 REFERENCING

As the referencing cost doesn't depend on the task *i.e.* the loss function, the results are evaluated on the model trained by approximation task and the results can be found in Table 7. The results reveal that although SincKAN has much more parameters than KAN and ChebyKAN, SincKAN is faster when referencing. Note that the referencing is slower than training because we compile the training procedure by JAX Bradbury et al. (2018).

Table 8: Computational cost for train PIKANs

| Function name | Network | Size | Training rate (iter/sec) | Parameters |
|---|---|---|---|---|
| boundary layer[‡] | MLP | $10 \times 100$ | $6.47 \times 10^2$ | 81101 |
| | modified MLP | $10 \times 100$ | $3.55 \times 10^2$ | 81501 |
| | KAN | $8 \times 8$ | $1.33 \times 10^3$ | 160 |
| | ChebyKAN | $40 \times 40$ | $1.89 \times 10^3$ | 3280 |
| | SincKAN | $8 \times 8 \times 1$ | $1.27 \times 10^3$ | 194 |
| *perturbed*[‡] | MLP | $10 \times 100$ | $6.85 \times 10^2$ | 81101 |
| | modified MLP | $10 \times 100$ | $3.59 \times 10^2$ | 81501 |
| | KAN | $8 \times 8$ | $1.27 \times 10^3$ | 160 |
| | ChebyKAN | $40 \times 40$ | $1.07 \times 10^3$ | 3280 |
| | SincKAN | $8 \times 8 \times 1$ | $1.25 \times 10^3$ | 194 |
| *nonlinear*[‡] | MLP | $10 \times 100$ | $4.62 \times 10^2$ | 81101 |
| | modified MLP | $10 \times 100$ | $3.07 \times 10^2$ | 81501 |
| | KAN | $8 \times 8$ | $1.56 \times 10^3$ | 160 |
| | ChebyKAN | $40 \times 40$ | $1.53 \times 10^3$ | 3280 |
| | SincKAN | $8 \times 4 \times 1$ | $1.54 \times 10^3$ | 130 |
| *bl-2d*[‡] | MLP | $10 \times 100$ | $2.39 \times 10^2$ | 81201 |
| | modified MLP | $10 \times 100$ | $1.96 \times 10^2$ | 81801 |
| | KAN | $8 \times 8$ | $3.46 \times 10^2$ | 240 |
| | ChebyKAN | $40 \times 40$ | $4.95 \times 10^2$ | 4920 |
| | SincKAN | $8 \times 20 \times 1$ | $2.97 \times 10^2$ | 570 |
| *ns-tg*[†] | MLP | $10 \times 100$ | $1.71 \times 10^2$ | 81503 |
| | modified MLP | $10 \times 100$ | $1.51 \times 10^2$ | 82303 |
| | KAN | $8 \times 8$ | $2.55 \times 10^2$ | 480 |
| | ChebyKAN | $40 \times 40$ | $3.83 \times 10^3$ | 9840 |
| | SincKAN | $8 \times 8 \times 1$ | $2.77 \times 10^2$ | 550 |

# I  RELATIONSHIP BETWEEN DEGREE AND SIZE OF DATA

In Sinc numerical methods, the number of the sampled points is equal to the degree because each degree requires a corresponding value $f(jh)$ at the point $jh$ *i.e.* $N_{degree} = N_{points}$. However, in SincKAN, $N_{degree} = N_{points}$ is impractical, and also not necessary because Sinc numerical methods can be regarded as a single-layer representation while our SincKAN is a multi-layer representation where the multi-layer representation has an exponentially increasing capability with depth Yarotsky (2017). To explore the relationship between degree and size of data, we train our SincKAN with different $N_{degree}$ and $N_{points}$. In this experiment, we train our SincKAN on *spectral-bias* function on $N = 8, 16, 32, 64, 100, 300$ and $N_{points} = 100, 500, 1000, 5000, 10000$ with the inverse decay $\{h_i\}_{i=1}^{M}$ in $M = 6$ and $h_0 = 7.0$. Moreover, we set the batch size $N_{batch} = N_{points}/4$ to adapt to the changing of $N_{points}$. The results are shown in Table 9 and Fig. 9. Additionally, Fig. 9(a) shows that our neural scaling law is RMSE $\propto G^{-4}$ compared to the best scaling law RMSE $\propto G^{-3}$ claimed in KAN Liu et al. (2024b).

# J  UPDATE OF GRIDS

The interpolation of every $\phi$ is an important composition of KANs. However, the degree of the interpolation methods is always determined empirically. Table 9 shows that larger degree doesn't mean better accuracy. One may argue that large model may cause over-parametrization. Herein, we

Table 9: RMSE for different degree and $N_{points}$

| degree $\setminus N_{points}$ | 100 | 500 | 1,000 | 5,000 | 10,000 |
|---|---|---|---|---|---|
| 8 | $2.60e\text{-}1 \pm 2.76e\text{-}5$ | $1.16e\text{-}1 \pm 6.34e\text{-}6$ | $8.23e\text{-}2 \pm 4.54e\text{-}6$ | $6.70e\text{-}2 \pm 3.91e\text{-}3$ | $6.81e\text{-}2 \pm 4.51e\text{-}3$ |
| 16 | $1.01e\text{-}2 \pm 1.47e\text{-}2$ | $1.30e\text{-}3 \pm 9.42e\text{-}4$ | $1.04e\text{-}3 \pm 3.74e\text{-}4$ | $1.62e\text{-}3 \pm 2.48e\text{-}4$ | $9.57e\text{-}4 \pm 4.35e\text{-}4$ |
| 32 | ${\color{red}3.63e\text{-}5 \pm 5.79e\text{-}5}$ | $2.69e\text{-}4 \pm 2.27e\text{-}4$ | $4.72e\text{-}4 \pm 2.75e\text{-}4$ | $2.15e\text{-}3 \pm 1.57e\text{-}3$ | $3.08e\text{-}3 \pm 7.14e\text{-}4$ |
| 64 | $1.29e\text{-}3 \pm 2.21e\text{-}3$ | $5.60e\text{-}4 \pm 6.42e\text{-}4$ | $2.74e\text{-}3 \pm 4.01e\text{-}3$ | $1.83e\text{-}3 \pm 8.49e\text{-}4$ | $2.13e\text{-}3 \pm 9.05e\text{-}4$ |
| 100 | $7.66e\text{-}5 \pm 1.16e\text{-}4$ | $3.79e\text{-}4 \pm 6.27e\text{-}4$ | $4.24e\text{-}4 \pm 7.42e\text{-}5$ | $2.19e\text{-}3 \pm 1.62e\text{-}3$ | $2.64e\text{-}3 \pm 1.76e\text{-}3$ |
| 300 | $3.73e\text{-}4 \pm 5.62e\text{-}4$ | $7.30e\text{-}5 \pm 4.25e\text{-}5$ | $7.54e\text{-}4 \pm 8.73e\text{-}4$ | $2.21e\text{-}3 \pm 1.04e\text{-}3$ | $2.18e\text{-}3 \pm 1.12e\text{-}3$ |

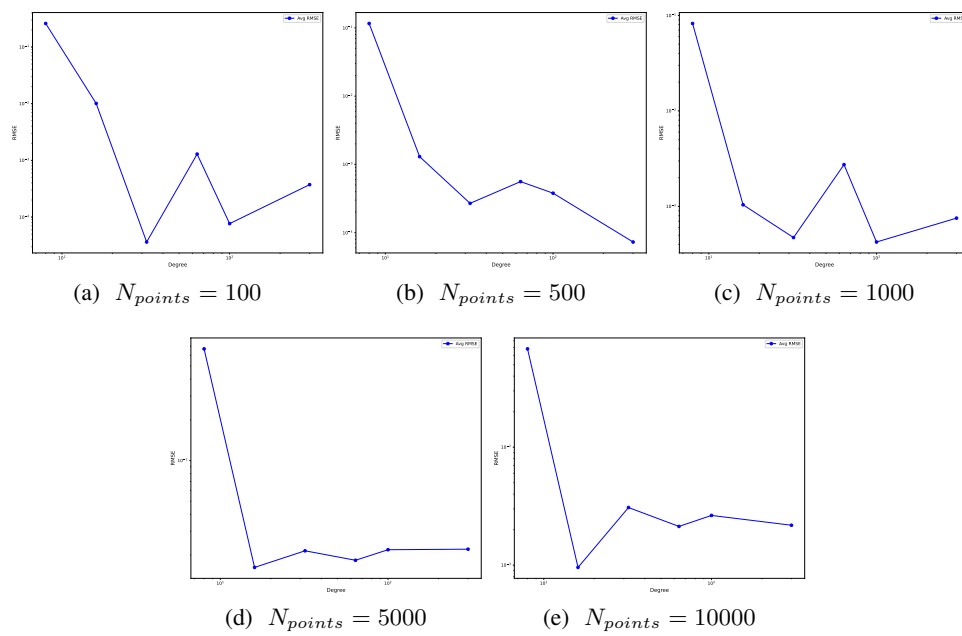

(a) $N_{points} = 100$      (b) $N_{points} = 500$      (c) $N_{points} = 1000$

(d) $N_{points} = 5000$      (e) $N_{points} = 10000$

Figure 9: Figures of different $N_{ponits}$ with increasing degree.

propose the algorithm Algorithm 1) of grid extension so that SincKAN can start from small degree and end at large degree.

---

**Algorithm 1** Update grids

---

**Input:** $M > N > 0, \{c_i\}_{i=-N}^{N}$,

**Output:** $\{c_i'\}_{i=-M}^{M}$

1: **for all** $i = -M, -N + 1, \cdots, M$ **do**

2:     **if** $i \in [-N, N]$ **then**

3:         $c_i' = c_i$;

4:     **else**

5:         $c_i = 0$

6:     **end if**

7: **end for**

---

We test this method in the same hyperparameter in Appendix I on *spectral-basis* and *bl* functions. For the first experiment, we set the constant degree $N_{degree} = 100$ . For variable degree, we begin with $N_{degree} = 8$ and update it by $[8, 8, 8, 8, 8, 8, 8, 8, 8, 8, 4, 4, 4]$ for 13 times to $N_{degree} = 100$. We train the constant degree for $1.4 \times 10^6$ iterations. For variable degree, we train each degree $10^5$ iterations, so the total number of training iterations is also $1.4 \times 10^6$, the results of update grids are shown in Fig. 10. We realized that there may have over-fitting problem in $N_{degree} = 100$, thus for the second experiment, we set the constant degree $N_{degree} = 96$ . For variable degree, we begin

with $N_{degree} = 8$ and update it by $[8, 8, 8, 8, 8, 8, 8, 8, 8, 8, 4, 4]$ for 13 times to $N_{degree} = 96$. We train the constant degree for $1.3 \times 10^6$ iterations. For variable degree, we train each degree $10^5$ iterations, so the total number of training iterations is also $1.3 \times 10^6$, the results of update grids are shown in Fig. 11.

Note that, because of the cost of changing the basis, using variable degree is slightly faster than using constant degree. The cost is shown in Table 10.

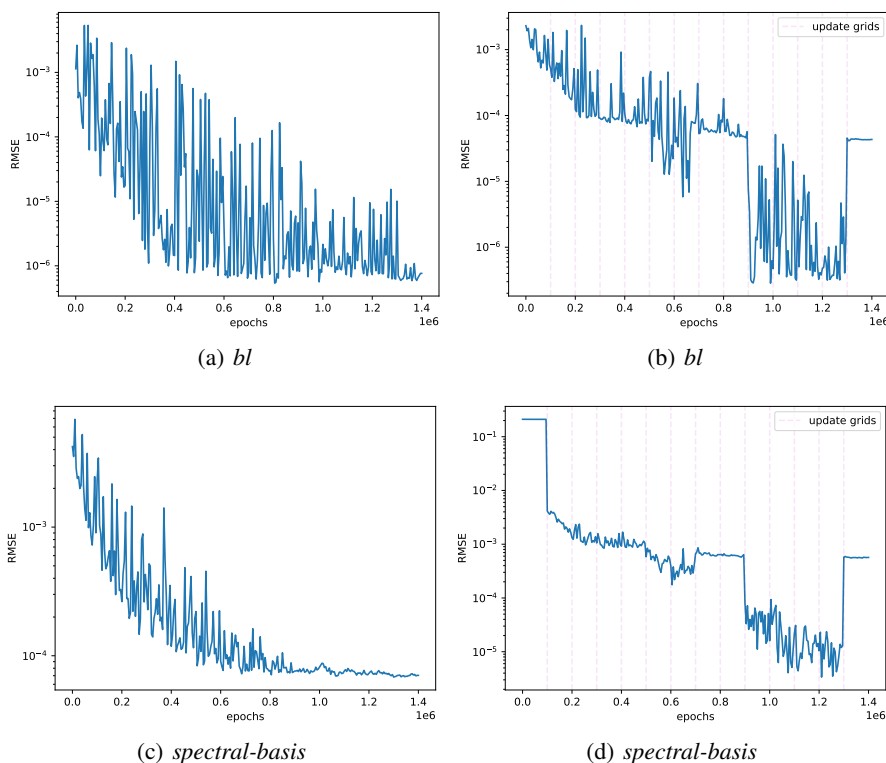

Figure 10: For *bl* and *spectral-basis*, Fig. 10(a) and Fig. 10(c) correspondingly shows the RMSE for the constant degree $N_{degree} = 96$; Fig. 10(b) and Fig. 10(d) correspondingly shows the RMSE for the variable degree

Table 10: Computational cost for update of grids

| | | |
|---|---|---|
| constant degree | *spectral-bias* | $7.82 \times 10^2$ |
| variable degree | *spectral-bias* | $7.82 \times 10^2$ |
| constant degree | *bl* | $7.67 \times 10^2$ |
| variable degree | *bl* | $7.73 \times 10^2$ |

## K    RESULTS OF SELECTED H

Table 11 and Table 12 show the results of selected $\{h_i\}$ in details. However, there are so many hyperparameters that may be adjusted when $h_i$ is larger. For example, for the large $h$ on fine grids, we argue that $N_{degree} = 100$ may not exploit the capability fully. Thus, we conducted an extra experiment with 5000 grid points, $\{h_i\} = \{1/10, 1/100, 1/1000\}$, and $N_{degree} = 500$. For sin-low, the RMSE is 7.92e-4 ± 4.21e-4, and for the sin-high, the RMSE is 2.32e-3 ± 2.74e-4. It shows that for sin-high, the SincKAN can obtain a more accurate result if we further tune the hyperparameters.

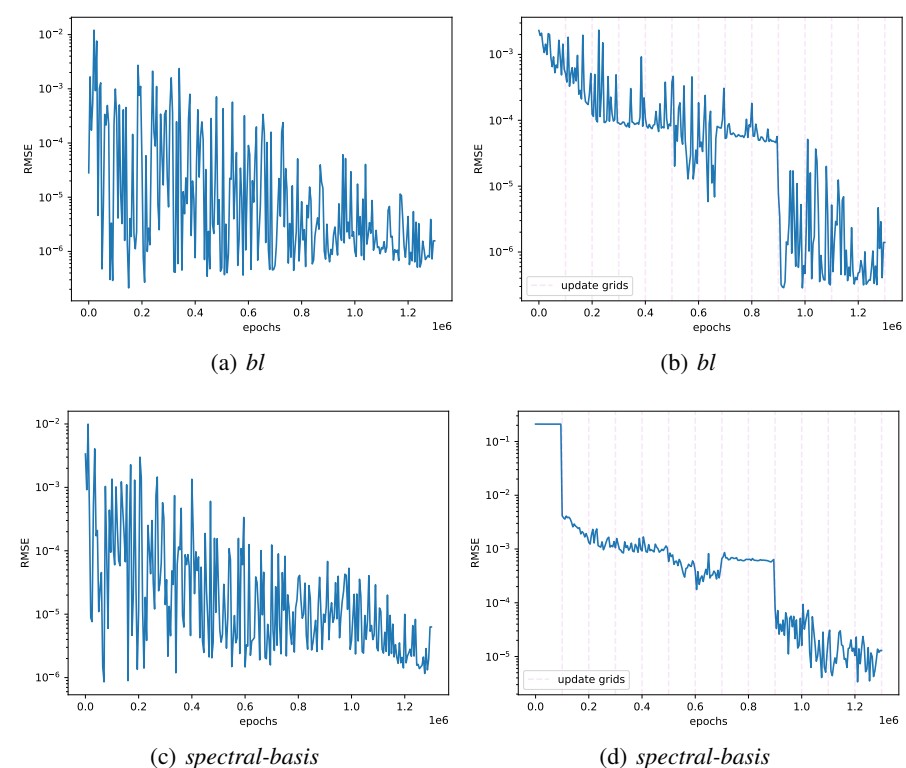

Figure 11: For *bl* and *spectral-basis*, Fig. 11(a) and Fig. 11(c) correspondingly shows the RMSE for the constant degree $N_{degree} = 96$; Fig. 11(b) and Fig. 11(d) correspondingly shows the RMSE for the variable degree

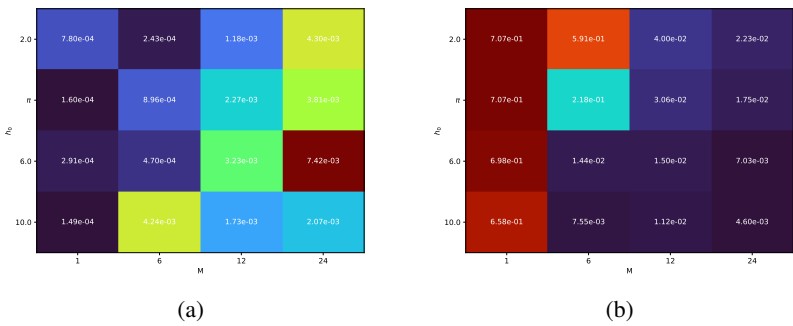

Figure 12: Fig. 12(a) shows the inverse decay approach on sin-low function; Fig. 12(b) shows the inverse decay approach on sin-high function.

## L  OSCILLATIONS OF SINCKAN

We conducted the experiments on convection-diffusion equations (Eq. (50)) with $h_0 = 2.0, 10.0$, $N = 8, 100$, and $M = 1, 6$. Except $h_0 = 2.0$ and $N = 8$, the inaccuracy of derivatives makes SincKAN unstable with the loss diverging. We choose some figures plotted in Fig. 14 to show the oscillations that limit the improvement of SincKANs.

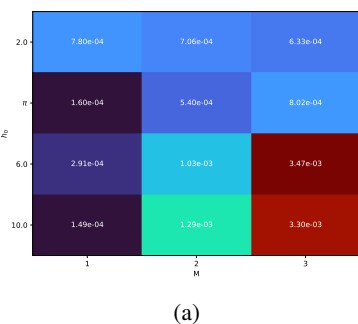 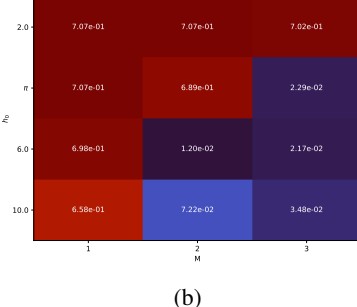

$$(a) \qquad\qquad (b)$$

Figure 13: Fig. 13(a) shows the exponential decay approach on sin-low function; Fig. 13(b) shows the exponential decay approach on sin-high function.

Table 11: RMSE for different $\{h_i\}$ with inverse decay

| Function name | $h_0 \setminus$ M | 1 | 6 | 12 | 24 |
|---|---|---|---|---|---|
| sin-low | 2.0 | $7.80e\text{-}4 \pm 7.96e\text{-}4$ | $2.43e\text{-}4 \pm 1.55e\text{-}4$ | $1.18e\text{-}3 \pm 2.38e\text{-}4$ | $4.30e\text{-}3 \pm 1.74e\text{-}3$ |
| | $\pi$ | $1.60e\text{-}4 \pm 6.66e\text{-}5$ | $8.96e\text{-}4 \pm 5.91e\text{-}4$ | $2.27e\text{-}3 \pm 8.53e\text{-}4$ | $3.81e\text{-}3 \pm 2.47e\text{-}3$ |
| | 6.0 | $2.91e\text{-}4 \pm 1.22e\text{-}4$ | $4.70e\text{-}4 \pm 1.88e\text{-}4$ | $3.23e\text{-}3 \pm 8.47e\text{-}4$ | $7.42e\text{-}3 \pm 7.40e\text{-}3$ |
| | 10.0 | $1.49e\text{-}4 \pm 8.74e\text{-}5$ | $4.24e\text{-}3 \pm 3.18e\text{-}3$ | $1.73e\text{-}3 \pm 1.08e\text{-}3$ | $2.07e\text{-}3 \pm 8.84e\text{-}4$ |
| sin-high | 2.0 | $7.07e\text{-}1 \pm 6.70e\text{-}6$ | $5.91e\text{-}1 \pm 6.32e\text{-}3$ | $4.00e\text{-}2 \pm 1.24e\text{-}3$ | $2.23e\text{-}2 \pm 1.38e\text{-}3$ |
| | $\pi$ | $7.07e\text{-}1 \pm 7.94e\text{-}6$ | $2.18e\text{-}1 \pm 1.88e\text{-}2$ | $3.06e\text{-}2 \pm 2.80e\text{-}3$ | $1.75e\text{-}2 \pm 2.44e\text{-}3$ |
| | 6.0 | $6.98e\text{-}1 \pm 4.88e\text{-}3$ | $1.44e\text{-}2 \pm 1.34e\text{-}3$ | $1.50e\text{-}2 \pm 8.68e\text{-}4$ | $7.03e\text{-}3 \pm 1.95e\text{-}3$ |
| | 10.0 | $6.58e\text{-}1 \pm 3.32e\text{-}3$ | $7.55e\text{-}3 \pm 1.73e\text{-}3$ | $1.12e\text{-}2 \pm 2.75e\text{-}3$ | $4.60e\text{-}3 \pm 3.70e\text{-}4$ |

## M  METRICS

In this paper, we use two metrics. For interpolation, we inherit the RMSE metric from KAN Liu et al. (2024b), the formula is :

$$\text{RMSE} = \sqrt{\frac{1}{N} \sum_{i=1}^{N} (y_i - \hat{y}_i)^2};\tag{66}$$

for PIKANs, we utilize the relative L2 error which is the most common metric used in PINNs:

$$\text{RelativeL2} = \frac{\|\boldsymbol{y} - \hat{\boldsymbol{y}}\|_2}{\|\boldsymbol{y}\|_2},\tag{67}$$

where $y$ is the target value, and $\hat{y}$ is the predicted value

Table 12: RMSE for different $\{h_i\}$ with exponential decay

| Function name | $h_0 \setminus M$ | 1 | 2 | 3 |
|---|---|---|---|---|
| sin-low | 2.0 | $7.80e\text{-}4 \pm 7.96e\text{-}4$ | $7.06e\text{-}4 \pm 2.54e\text{-}4$ | $6.33e\text{-}4 \pm 1.38e\text{-}4$ |
| | $\pi$ | $1.60e\text{-}4 \pm 6.66e\text{-}5$ | $5.40e\text{-}4 \pm 2.05e\text{-}4$ | $8.02e\text{-}4 \pm 6.58e\text{-}5$ |
| | 6.0 | $2.91e\text{-}4 \pm 1.22e\text{-}4$ | $1.03e\text{-}3 \pm 3.78e\text{-}4$ | $3.47e\text{-}3 \pm 3.31e\text{-}4$ |
| | 10.0 | $\textcolor{red}{1.49e\text{-}4 \pm 8.74e\text{-}5}$ | $1.29e\text{-}3 \pm 1.74e\text{-}4$ | $3.30e\text{-}3 \pm 2.80e\text{-}4$ |
| sin-high | 2.0 | $7.07e\text{-}1 \pm 6.70e\text{-}6$ | $7.07e\text{-}1 \pm 1.42e\text{-}5$ | $7.02e\text{-}1 \pm 2.84e\text{-}3$ |
| | $\pi$ | $7.07e\text{-}1 \pm 7.94e\text{-}6$ | $6.89e\text{-}1 \pm 4.82e\text{-}3$ | $2.29e\text{-}2 \pm 1.76e\text{-}3$ |
| | 6.0 | $6.98e\text{-}1 \pm 4.88e\text{-}3$ | $\textcolor{red}{1.20e\text{-}2 \pm 1.35e\text{-}3}$ | $2.17e\text{-}2 \pm 6.67e\text{-}4$ |
| | 10.0 | $6.58e\text{-}1 \pm 3.32e\text{-}3$ | $7.22e\text{-}2 \pm 8.12e\text{-}3$ | $3.48e\text{-}2 \pm 2.38e\text{-}3$ |

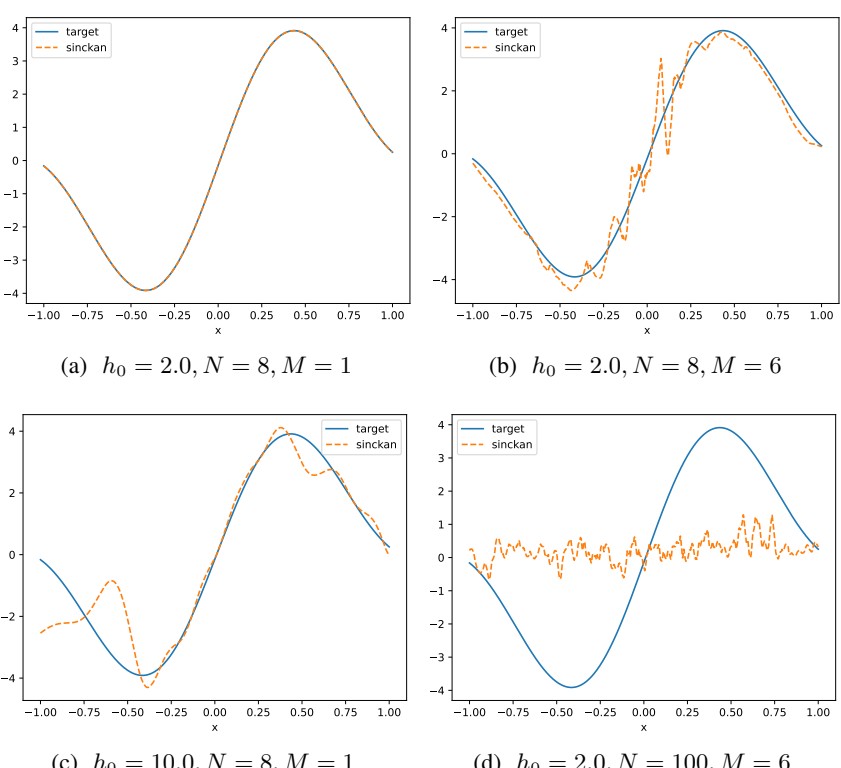

(a) $h_0 = 2.0, N = 8, M = 1$

(b) $h_0 = 2.0, N = 8, M = 6$

(c) $h_0 = 10.0, N = 8, M = 1$

(d) $h_0 = 2.0, N = 100, M = 6$

Figure 14: Fig. 14(a) solve Eq. (50) accurately. However, SincKANs have oscillations after either increasing $M$ (Fig. 14(b)) or increasing $h_0$. Fig. 14(d) shows that with the same hyperparameters used in approximation, SincKAN becomes extremely inaccurate due to the violent oscillations.

