# OpenReview forum: "Sinc Kolmogorov-Arnold Network and Its Applications on Physics-informed Neural Networks"
_ICLR.cc/2025/Conference — Submitted to ICLR 2025_

### Official Review · Reviewer_gzak · 2024-11-01

**Soundness:** 3
**Presentation:** 2
**Contribution:** 2
**Rating:** 6
**Confidence:** 3

**Summary:**

The work proposes using Sinc interpolations replace the splines used within recently popularised Kolomogorov Arnold Networks (KANs). The work outlines foundational background required of Sinc interpolation and demonstrates some nice properties of Sinc interpolation on some specific benchmarks. It then outlines certain properties that must hold for effective Sinc interpolation and incorporates these within its KAN framework. Benchmarks are performed against against baselines comprising of two different multilayer perceptron architectures, and two other KAN architectures. These experiments encompass a range of function approximation tasks for manufactured functions, and physics-informed tasks.

**Strengths:**

1. KANs are recent and might have much scope for optimisation. The interpolation scheme in KANs seems to be an important point to consider in practical use, so it is good that the authors draw attention to this.
2. Given the recency of KANs, the study provides more independent comparisons between KANs and MLPs which might be of interest to the wider community.
3. The authors do not straightforwardly substitute Sinc interpolants into the KAN architecture, but properly consider its principled application in terms of optimal step sizes, required exponential decay of the approximated function and how to circumvent this etc.

**Weaknesses:**

1. The manufactured solutions used in the learning for approximation secion (3.1) are arbitrary and seemingly not very relevant to real world applications. They are both manufactured, and all are one dimensional.
2. While the performance of SincKANs is good in physics-informed settings, it feels like some scrutiny of the results is missing. There are many reasons why a physics-informed problem might fail to converge. It might not be that the reason for better performance in a physics-informed problem is a better architecture.

**Questions:**

Question:
1. Can't one use a Sinc interpolator directly for all the function approximation tasks?

Sugestions:
1. The experiments of section 3.1 do not seem so relevant for practical applications. It seems that what might be most interesting would be to take a selection of complicated PDE/ODE data (using any off the shelf differential equation solver) and use these instead. This would demonstrate the value of SincKANs in practical circumstances, and in situations of more than 1 dimension. As it stands, these benchmarks feel artificial and irrelevant for real world application.
2. More scrutiny of the physics-informed losses would be beneficial. Some plots of solutions and errors across the poorer performing methods might help understand why they are performing badly. Is it that boundary conditions are not being adhered to? Maybe there are regions of high PDE loss in the resulting solution? Perhaps small changes (e.g. weighing boundary conditions effectively) might lead to improved performance.

---

> ### Author Response · Authors · 2024-11-19
>
> # Can't one use a Sinc interpolator directly for all the function approximation tasks?
>
> Sinc interpolator with a fixed $h$ can be used for all the function interpolation tasks, but it won't guarantee the theorem 1 and 2 if $h$ is not the optimal one. In our SincKAN, as the optimal $h$ is unknown, the performance of a single Sinc interpolator is poor so we change to use several Sinc interpolators. Furthermore, several Sinc interpolators are theoretically more accurate than a singular interpolator, as we discussed on line 243 in our manuscript.
>
> # The experiments of section 3.1 do not seem so relevant for practical applications. It seems that what might be most interesting would be to take a selection of complicated PDE/ODE data (using any off the shelf differential equation solver) and use these instead. This would demonstrate the value of SincKANs in practical circumstances, and in situations of more than 1 dimension. As it stands, these benchmarks feel artificial and irrelevant for real world application.
>
> The experiments of section 3.1 are not manufactured and impractical, *sin-low* and *sin-high* are from [RICHARDSON et.al (2011)](https://epubs.siam.org/doi/ref/10.1137/110825947), *double-exponential* and *multi-sqrt* is from [Sugihara et.al (2003)](https://www.sciencedirect.com/science/article/pii/S0377042703008380),  *piece-wise* is from ChebyKAN, *spectral-bias* is from [Rahaman et.al (2019)](https://arxiv.org/abs/1806.08734).
>
> We also did experiments on high-dimensional functions,
>
> ### For approximation
> let's consider the function:$f(\boldsymbol{x})=e^{-\alpha ||\boldsymbol{x}||^2_2}, \boldsymbol{x}\in [-1,1]^d$ from [DAS-PINNs](https://arxiv.org/abs/2112.14038). Here we choose $\alpha=0.001, d=100.$
>
> The results are:
> | Model   | RMSE |
> |---------|----------------|
> |  MLP     | 5.511e-03    |
> | SincKAN | 2.867e-03    |
>
> ### For PDE:
> We consider two different problems: the fractional PDEs, and high-dimensional Poisson.
> For fractional PDEs, we utilize [fPINN](https://arxiv.org/abs/1811.08967) on the spatial fractional order problem:
> \begin{equation}
>     -\Delta^su+\gamma u=f, \quad \boldsymbol{x}\in (-1,1)^d
> \end{equation}
> if $f=1+\gamma u$, the exact solution is
> \begin{equation}
>     u(x)=\frac{2^{-2s}\Gamma(d/2)}{\Gamma(d/2+s)\Gamma(1+s)}(1-\|x\|^2)^s,
> \end{equation}
> where $s\in \mathbb{R}_+$ is a hyperparameter. In our experiments, $s=0.95,d=1$.
>
> The results are:
> | Model   | Relative error|
> |---------|----------------|
> |  MLP     | 7.32e-02       |
> | SincKAN | 7.28e-03       |
>
> For high-dimensional Poisson equations. we consider
> \begin{equation}
>     \Delta u =f, \quad \boldsymbol{x}\in [0,1]^d.
> \end{equation}
> where $f=2\alpha(2\alpha ||\boldsymbol{x}||^2_2-d) e^{-\alpha ||\boldsymbol{x}||_2^2}$, the exact solution is $u(\boldsymbol{x})=e^{-\alpha ||\boldsymbol{x}||^2_2}$. In our experiments, $d=100, \alpha=10/d$.
>
> The results are:
> | Model   | Relative error|
> |---------|----------------|
> |  MLP     | 7.58e-02      |
> | SincKAN | 3.49e-02     |
>
> # More scrutiny of the physics-informed losses would be beneficial. Some plots of solutions and errors across the poorer performing methods might help understand why they are performing badly. Is it that boundary conditions are not being adhered to? Maybe there are regions of high PDE loss in the resulting solution? Perhaps small changes (e.g. weighing boundary conditions effectively) might lead to improved performance.
>
> We have already plotted the solutions to the poor performance in Figure 4 (c). The figure shows that the boundary condition is strictly obeyed for every network because we use weight=100 for boundary loss and weight=1 for residual loss. Besides, we did experiments on larger weights of boundary conditions to have a more strict boundary condition: we keep the weight of residual loss, and weight=1000 for boundary loss in the poor performance experiments i.e. $\epsilon=1000$ in Table 3. Additionally, we release the boundary condition by setting weight=10 to further explore the performance. Here are the results:
> | weight | $10^1$ |$10^2$ (from our manuscript) | $10^3$|
> |---------|----------------|----------------|----------------|
> |  MLP     |  1.76e01    | 9.87 $\pm$ 8.70 | 2.32e01

---

> > ### Comment · Reviewer_gzak · 2024-11-21
> >
> > I personally feel that there should be more 2D and higher dimensional tasks for the function approximation. One of the points of the Kolmogorov-Arnold representation theorem is to approximate multivariate functions with univariate functions. Sticking to univariate benchmarks seems to both i) to ignore one of the fundamental points of KANs and ii) limits these cutting edge models to situations where generally we do have many practical machine learning difficulties (1D function approximation).
> >
> > However, I do appreciate that the authors have addressed my questions. I overall think that avenues to improve KANs and investigate their performance is of interest to the wider community, so I will increase my score to a 6.

---

> > > ### Author Response · Authors · 2024-11-26
> > >
> > > Thanks for your comment about the approximation. We thought that 1D experiments were enough for approximation because we can use high-dimensional PDE problems to indirectly show the capability of approximating higher-dimensional functions of SincKAN.
> > >
> > > After our discussion, we agreed that more high-dimensional approximation experiments should be added. Thus, we added some extra experiments for 2D and 4D functions. You can find them in Table 1 and Table 5.

---

> > > > ### Comment · Reviewer_gzak · 2024-11-27
> > > >
> > > > The addition of higher-dimensional benchmarks is helpful.
> > > >
> > > > However, while the authors have added more benchmarks and addressed my concerns of there not being high-dimensional problems, the original weakness 1 of the original review persists
> > > >
> > > > ```
> > > > The manufactured solutions used in the learning for approximation secion (3.1) are arbitrary and seemingly not very relevant to real world applications.
> > > > ```
> > > >
> > > > The benchmarks all have closed-form solutions. Many are in terms of sins and exponentials, that it stands to reason are well approximated by Sincs. These are not the sorts of functions encountered in practical modelling. So my concerns about real world applicability persist and so I am hesitant to increase my score further.

---

> ### Author Response · Authors · 2024-11-27
>
> # Many are in terms of sins and exponentials, that it stands to reason are well approximated by Sincs.
>
> 1) interpolation is an important composition of KANs, but not all.
>
> 2) Sinc function has different features with sins and exponentials, it is more like a 'quasi' delta function rather than sins and exponentials. For example let's interpolate $\sin(2\pi x), x\in[0,1]$, by @chebfun and @sincfun tools in MATLAB, like what we did in Figure1. The results are:
> ```
> degree: 1.00e+01,sinc: 9.98e-01, cheby: 8.58e-06, cubic: 2.24e-03,
>
> degree: 2.00e+01,sinc: 8.95e-01, cheby: 7.71e-16, cubic: 4.37e-05,
>
> degree: 5.00e+01,sinc: 1.73e-01, cheby: 3.72e-16, cubic: 4.99e-07,
>
> degree: 8.00e+01,sinc: 1.65e-02, cheby: 4.59e-16, cubic: 6.83e-08,
>
> degree: 1.00e+02,sinc: 2.95e-03, cheby: 4.04e-16, cubic: 2.73e-08,
>
> degree: 1.50e+02,sinc: 2.94e-05, cheby: 5.14e-16, cubic: 5.27e-09,
>
> degree: 2.00e+02,sinc: 2.18e-07, cheby: 5.07e-16, cubic: 1.65e-09,
>
> degree: 2.50e+02,sinc: 1.33e-09, cheby: 5.43e-16, cubic: 6.74e-10,
>
> degree: 3.00e+02,sinc: 7.10e-12, cheby: 4.23e-16, cubic: 3.24e-10,
>
> degree: 4.00e+02,sinc: 9.84e-16, cheby: 3.96e-16, cubic: 1.02e-10,
>
> degree: 5.00e+02,sinc: 1.03e-15, cheby: 5.67e-16, cubic: 4.17e-11,
> ```
> Herein, the cubic and Chebyshev are better than Sinc for interpolating $\sin$ if degree<300.
>
> We conducted experiments on *lpmv*, *ellipj*, *sph-harm* from KAN, the results are shown in the new revision (Table 1 and 5).

---

### Official Review · Reviewer_wJoX · 2024-11-03

**Soundness:** 3
**Presentation:** 2
**Contribution:** 2
**Rating:** 6
**Confidence:** 2

**Summary:**

The paper builds upon recently proposed architecture KANs that are meant to be more parameter efficient than MLPs and provide additional benefits like interpretability. The authors replace the splines in KANs with sinc to propose SincKAN. Furthermore, the paper presents experiments in PINNs (using KANs) to demonstrate potential benefits of SincKAN.

**Strengths:**

1. Applying sinc to replace splines in KAN is a novel approach. Additionally, the authors address the practical challenges of selecting step sizes using multiple step sizes.

**Weaknesses:**

1. The PDEs selected for the experiments are generally unsuitable to demonstrate the benefits of PIKANs. The chosen problems are extremely small and these problems can be solved using any numerical PDE solver. Since PINNs / PIKANs don't provide any practical advantage in small problems, it would be interesting to see if the benefits proposed here can be demonstrated in real problems.

**Questions:**

1. For MLP architectures, how were the architectures chosen? Since the problems are small and toy-ish, over-parameterization may cause a higher final loss.

---

> ### Author Response · Authors · 2024-11-14
>
> # The PDEs selected for the experiments are generally unsuitable to demonstrate the benefits of PIKANs. The chosen problems are extremely small and these problems can be solved using any numerical PDE solver. Since PINNs / PIKANs don't provide any practical advantage in small problems, it would be interesting to see if the benefits proposed here can be demonstrated in real problems.
>
> Dear reviewer, this is a quick comment on the chosen experiments.
>
> Our motivation of proposing the SincKAN is to solve the boundary layer problems and singularity problems which have difficulties in PINNs. Those problems represent a kind of problems called perturbed problems that have wild applications e.g. chemical reactors: [Turian et al. (1973)](https://www.sciencedirect.com/science/article/pii/0009250973850468), and fluid dynamics: [Gordon et al. (2000)](https://asmedigitalcollection.asme.org/dynamicsystems/article-abstract/122/4/699/445484/Modeling-Realization-and-Simulation-of-Thermo). Additionally, although numerical PDE solvers can solve perturbed problems, those approaches often require expert knowledge and specific tricks.
>
> To further show the advantages of SincKAN, we are going to add more numerical examples, including the high-dimensional PDEs, which the traditional numerical methods have the curse of dimensionality, and the fractional PDEs which the traditional numerical methods still face challenges due to their non-locality and singularity. We believe those problems can show the benefits of PIKANs and convince you to recognize our contributions.

---

> ### Author Response · Authors · 2024-11-20
>
> # For MLP architectures, how were the architectures chosen? Since the problems are small and toy-ish, over-parameterization may cause a higher final loss.
>
> Thanks for the reminder, our hyperparameters are from the [NSFNet](https://arxiv.org/abs/2003.06496), the most powerful model for solving the high-resolution dataset [Johns Hopkins Turbulence Database](https://turbulence.pha.jhu.edu/). After your question, we think we should test some small MLP networks. To show the performance of small sizes of MLP, we run experiments on the 1D and 2D boundary layer problem with different sizes.
>
> ## For 1D boundary layer problems
> | size |  $\epsilon=100$ | $\epsilon=1000$
> |--|--|--|
> $2\times 50$ | 6.62e-02 |1.70e+01
> $2\times 100$ | 1.42e-01 |  1.70e+01
> $3\times 50$ | 7.66e-04 |2.82e+01
> $3\times 100$ | 4.25e-04 |  2.42e+01
> $5\times 50$ | 2.44e-05 |1.49e-02
> $5\times 100$| 5.55e-04 | 2.48e-02
>
>
> ## For 2D boundary layer problems
>
> From the above results of 1D boundary layer problems, we found that the competitive size is $5\times 50$ and $5\times 100$,, herein, in 2D case, we only tested those two size, the results are
> | size |  $\epsilon=100$
> |--|--|
> $5\times 50$ | 8.15e-02
> $5\times 100$| 6.52e-02
>
> However, only in 1D case, the result is better than our SincKAN, otherwise, the results are still worse than SincKAN.
>
> # it would be interesting to see if the benefits proposed here can be demonstrated in real problems.
>
> We consider two different problems: the fractional PDEs, and high-dimensional Poisson.
>
> For fractional PDEs, we utilize [fPINN](https://arxiv.org/abs/1811.08967) on the spatial fractional order problem:
> \begin{equation}
>     -\Delta^su+\gamma u=f, \quad \boldsymbol{x}\in (-1,1)^d
> \end{equation}
> if $f=1+\gamma u$, the exact solution is
> \begin{equation}
>     u(x)=\frac{2^{-2s}\Gamma(d/2)}{\Gamma(d/2+s)\Gamma(1+s)}(1-\|x\|^2)^s,
> \end{equation}
> where $s\in \mathbb{R}_+$ is a hyperparameter. In our experiments, $s=0.95,d=1$.
>
> The results are:
> | Model   | Relative error|
> |---------|----------------|
> |  MLP     | 7.32e-02       |
> | SincKAN | 7.28e-03       |
>
> For high-dimensional Poisson equations. we consider
> \begin{equation}
>     \Delta u =f, \quad \boldsymbol{x}\in [0,1]^d.
> \end{equation}
> where $f=2\alpha(2\alpha ||\boldsymbol{x}||^2_2-d) e^{-\alpha ||\boldsymbol{x}||_2^2}$, the exact solution is $u(\boldsymbol{x})=e^{-\alpha ||\boldsymbol{x}||^2_2}$. In our experiments, $d=100, \alpha=10/d$.
>
> The results are:
> | Model   | Relative error|
> |---------|----------------|
> |  MLP     | 7.58e-02      |
> | SincKAN | 3.49e-02     |
>
> Finally, we want to clarify that our work is to explore the implementations of KANs in PINNs and propose a powerful network that is helpful in PINNs. We are interested in solving numerical PDEs rather than discovering physical laws (i.e. the interpretability). Herein we are focusing on whether KAN is more accurate than MLP as [KAN](https://arxiv.org/abs/2404.19756) claimed that can help PINNs solve PDEs, or which kind of version of KAN may be helpful to us.

---

> ### Author Response · Authors · 2024-11-24
> **Additional experiments**
>
> Dear reviewer wJoX,
>
> Reviewer mr4i asked about the experiments for larger $\epsilon$, so we conducted the experiments on $\epsilon=10^4$ for boundary layer problems and added them in Table 3. For your concerned questions, we also did experiments on small size MLP, the results are:
>
> | size | $\epsilon=10^4$|
> |--|--|
> |$5\times 50$| 12.33|
> | $5 \times 100$ | 9.25|

---

> ### Author Response · Authors · 2024-11-28
>
> Dear reviewer,
>
> Is there anything else that we haven't answered or haven't answered fully? We believe that we have answered the questions and weaknesses from you fully and comprehensively. And we argue that the rating misevaluates our manuscript so we kindly ask you to update your rating or have a further conservation with us.

---

### Official Review · Reviewer_FYYo · 2024-11-04

**Soundness:** 3
**Presentation:** 3
**Contribution:** 1
**Rating:** 5
**Confidence:** 5

**Summary:**

This paper proposes KANs with sinc function as basis to represent univariate function, to which they refer as sincKan. They first discuss about the sinc function and its convergence properties and then extend it such that it becomes a viable and practical workhorse in the context of KANs. Equation 14 clarifies their contributions, namely, 1) choosing step size without the knowledge of function f (by using different h values, larger and smaller than the optimal one), 2) coordinate transformation (a composition of transformation from finite to infinite domain and a normalizing transformation), and 3) enforcing exponential decay (by considering g-f instead of f with a linear g). First property manifests itself in use of ci,j and hj, second one is hidden in $\gamma^{-1}$, and the third property is $c1x+c2$. They use both function approximation and solution of PDEs (similar to PINNs, which enforces the solution as soft constraint in the loss function) as numerical experiments and show lower error in most cases compared to relevant baselines.

**Strengths:**

The paper and idea are well-written. It is clear what authors try to implement and they are also rigor in deriving equation 14, which is a foundation for their sincKAN network.
The choice of boundary layer example is smart and shows the good performance of sincKAN in the presence of singularities (specially pronounced in large epsilon (lower width of boundary layer or larger Reynolds, which is more complicated).

**Weaknesses:**

The contribution of this paper seems limited to me and I hope authors clarify this in the rebuttal period.
Using more advanced basis functions for KANs is a meaningful research direction but currently it seems to me no single method can be seen as the best solution for all PDEs. In fact sincKAN may have better performance in the presence of singularities but may not be the best option otherwise. In the current version, I can't derive a concrete conclusion as which version of KANs to use for what type of problems (within PDE domain or even beyond). In fact, in Table 2, sincKAN is not the best option in some cases.
In an analogy to more conventional deep learning papers, the use of Relu activation functions with CNN enable them to achieve very good performance: it helped with vanishing gradient problem, it helped with efficiency of implementations, and also empirical success (for instance AlexNet). sincKAN does not seem to be such an abrupt idea. The theorems provided do not really guarantee the performance (specially considering the fact that authors use a range of step sizes instead of the optimal value due to practical reasons).
In this vein, authors discuss how sincKAN can help with spectral bias, but not much discussion is done on other issues for PINNs e.g. issue for causality (which makes them problematic in chaotic PDEs--which by the way is good to add to examples), and many other issues (like plateauing loss, etc.).
Finally, in "vanilla" KANs paper there are some arguments on the advantage of using B-splines, which enable them the use of grid related tricks including grid extension and grid update tricks. In other variants also there are some interesting merits; it's hard to say sincKAN still has all this merits and in addition to them provide more robustness in the case of singularities.

**Questions:**

Can sincKAN help with some pathological issues of PINNs? Would be interesting to explore; see for example Wang, S., Teng, Y., and Perdikaris, P. Understanding and Mitigating Gradient Flow Pathologies in Physics- Informed Neural Networks. SIAM Journal on Scientific Computing, 43(5):A3055–A3081, 2021a. or Wang, S., Sankaran, S., and Perdikaris, P. Respecting causal- ity is all you need for training physics-informed neural networks. arXiv preprint arXiv:2203.07404, 2022a. or the more recent paper Challenges in Training PINNs: A Loss Landscape Perspective by Rathore et al.

I still don't see the real advantages of theorems 1 and 2, since the optimal value of step is not uses in sincKAN. Also, I advise authors formally add another theorem to Coordinate transformation section to prove for the composition, the theory still holds (it should be straightforward). Also, if a function needs to be Hardy, does that mean the guarantees are only valid for differentiable functions f? It would be good to discuss the implementations.

---

> ### Author Response · Authors · 2024-11-19
>
> # Can sincKAN help with some pathological issues of PINNs? Would be interesting to explore; see for example Wang, S., Teng, Y., and Perdikaris, P. Understanding and Mitigating Gradient Flow Pathologies in Physics- Informed Neural Networks. SIAM Journal on Scientific Computing, 43(5):A3055–A3081, 2021a. or Wang, S., Sankaran, S., and Perdikaris, P. Respecting causal- ity is all you need for training physics-informed neural networks. arXiv preprint arXiv:2203.07404, 2022a. or the more recent paper Challenges in Training PINNs: A Loss Landscape Perspective by Rathore et al.
>
> Thanks for your question about the exploration of SincKAN. As SincKAN is an alternative network of MLP, approaches to changing other compositions can be implemented in SincKAN.  Let's consider the first two papers.
>
> For your first paper, Wang, S (2021), proposed a novel network that has already
> been used in our experiments(Modified MLP) that can't be combined with SincKAN totally, and an algorithm called learning rate annealing algorithm. The algorithm can improve the performance to 1.99e-04 in *perturbed* compared with the result 1.88e-03 in Table 2.
>
> In the second paper Wang, S (2022), proposed a novel formula for the loss function, we implement this formula with SincKAN in our NS problem. The novel loss can improve the performance a little, *ns-tg-u*: 7.07e-04, *ns-tg-v*: 4.58e-04 compared with the results *ns-tg-u*: 6.51e-04, *ns-tg-v*: 1.34e-03 in Table 2.
>
> As for the pathological issues, the boundary layer problems we used in our experiments actually belong to this category because the high-frequency part causes the spectral bias, see [Wang, S When and why PINNs fail to train: A neural tangent kernel perspective](https://www.sciencedirect.com/science/article/pii/S002199912100663X). Herein, thanks to the capability of handling singularity, SincKAN can help solve pathological issues by itself or by combining with other powerful technologies.
>
> # I still don't see the real advantages of theorems 1 and 2, since the optimal value of step is not uses in sincKAN.
>
> Theorem 1 and 2 are for Sinc numerical methods, especially the Sinc interpolations. Those two theorems inspired us to use Sinc interpolation in KANs. However, because applying the Sinc interpolation is not a good idea as we discuss in our manuscript, we unfold this approach to meet the demands of machine learning. Thus, this theorem is our fundamental theorem for replacing the cubic interpolation in KANs. Furthermore, interpolation is one of the most important compositions in KANs but not all of KANs. Thus, Theorem 1 and 2 can't dominate SincKAN totally.
>
> # Also, if a function needs to be Hardy, does that mean the guarantees are only valid for differentiable functions f? It would be good to discuss the implementations.
>
> Herein, in our SincKAN, it is not only valid for differentiable functions f, for example, *piece-wise* and *spectral-bias* are discontinuous functions.

---

> > ### Comment · Reviewer_FYYo · 2024-11-20
> >
> > Thanks authors for starting the conversation. This is to acknowledge that I will carefully go over them (and other reviewers' comments) and if needed, will engage in further conversation/update my score.

---

> > ### Comment · Reviewer_FYYo · 2024-11-27
> >
> > Authors addressed my concern for the first comment; however the second and third ones are really not addressed and in fact it proves my point. But I appreciate the conversation.

---

> ### Author Response · Authors · 2024-11-19
>
> # Also, I advise authors formally add another theorem to Coordinate transformation section to prove for the composition, the theory still holds (it should be straightforward).
>
> Thanks for your valuable suggestion, we will add the additional theorem on the Appendix, here is the theorem:
>
> ## Theorem 3
>
> Suppose $f(x), x\in \mathbb{R}$, satisfies \textbf{Theorem 1 }, that is,
>
> (1) $f$ belongs to $H^1\left(\mathcal{D}_d\right)$, where $H^1$ is the Hardy space and $\mathcal{D}_d=\{z\in \mathbb{C} \ | \ \left|\Im z\right|<d\}$;
>
> (2) $f$ decays exponentially on the real line, that is, $|f(x)| \leq \alpha \exp (-\beta|x|), \ \forall x \in \mathbb{R}$.
>
> Then for given $\mu \geq 0, \sigma > 0$ and $\xi=\frac{x-\mu}{\sigma}$, we want to prove that $\tilde{f}(\xi)=f(\sigma\xi+\mu), \xi \in \mathbb{R}$ satisfies condition (1) and (2).
>
> (1) As $\mu$ and $\sigma$ are real number,
>
> $$f(x)\in H^1\left(\mathcal{D}_d\right) $$
>
> $$ \Leftrightarrow  \lim_{\varepsilon \rightarrow 0} \int_{\partial \mathcal{D}_d(\varepsilon)}|f(z)||\mathrm{d} z|<\infty $$
>
> $$  \Leftrightarrow\int_{-\infty}^{\infty}|f(x \pm id)|\mathrm{d} x<\infty $$
>
> For $\tilde{f}$
>
> $$   \int_{-\infty}^{\infty}|\tilde{f}(\tilde{x}\pm i\tilde{d})|\mathrm{d} \tilde{x}$$
>
> $$  =  \int_{-\infty}^{\infty}|f(\sigma \tilde{x}+\mu \pm i\sigma \tilde{d})|\mathrm{d} x $$
>
> $$  =  \int_{-\infty}^{\infty}|f(x^\prime \pm id^\prime)|\mathrm{d} x^\prime, \text{ where } x^\prime=\sigma \tilde{x} +\mu,d^\prime = \sigma \tilde{d}$$
>
> If $d^\prime < d$, i.e. $\tilde{d}<\frac{d}{\sigma}$, then $ \int_{-\infty}^{\infty}|f(x^\prime\pm id^\prime)|\mathrm{d} x^\prime < \infty \Rightarrow \tilde{f}(\xi)\in H^1\left(\mathcal{D}_{\tilde{d}}\right)$. As $\frac{d}{\sigma}>0$, $\tilde{d}$ exists, (1) is satisfied.
>
> (2) As $|\tilde{f}(\xi)| \leq \alpha \exp (-\beta|\sigma\xi+\mu|), \ \forall \xi \in \mathbb{R}$, if there exists $\tilde{\alpha},\tilde{\beta}>0$ such that $\alpha \exp (-\beta|\sigma\xi+\mu|) \leq \tilde{\alpha}\exp (-\tilde{\beta}|\xi|) $, then (2) is satisfied:
>
> $$ \alpha \exp (-\beta|\sigma\xi+\mu|) \leq \tilde{\alpha}\exp (-\tilde{\beta}|\xi|) $$
>
> $$ \Rightarrow \log \alpha -\beta|\sigma\xi+\mu| \leq \log \tilde{\alpha} -\tilde{\beta}|\xi| $$
>
> $$ \Rightarrow \tilde{\beta} \leq \frac{\log \frac{\tilde{\alpha}}{\alpha}}{|\xi|}+\beta\left|\sigma+\frac{\mu}{\xi}\right|, \text{ if } \xi\neq 0.$$
>
> Thus, if $\tilde{\alpha}>\alpha$, there exists a $\tilde{\beta}>0$ that satisfies the above inequality; on the other hand if $\xi=0$, obviously $\alpha \exp (-\beta|\sigma\xi+\mu|) \leq \alpha < \tilde{\alpha}$. Herein, $\tilde{\alpha}, \tilde{\beta}$ exists.
>
> In total, as $\tilde{f}$ guarantees conditions (1) and (2), $\tilde{f}$ satisfies Theorem 1.

---

> > ### Comment · Reviewer_FYYo · 2024-11-27
> >
> > Good addition. Thanks (hope it helps the value of the paper)

---

> ### Author Response · Authors · 2024-11-19
>
> # Using more advanced basis functions for KANs is a meaningful research direction but currently it seems to me no single method can be seen as the best solution for all PDEs. In fact sincKAN may have better performance in the presence of singularities but may not be the best option otherwise. In the current version, I can't derive a concrete conclusion as which version of KANs to use for what type of problems (within PDE domain or even beyond).
>
> In our opinion, there is no best choice for all PDEs. In traditional methods, there are finite difference methods, finite element methods, finite volume methods, spectral methods, etc. Every method has its advantages and disadvantages, for example, the spectral methods are faster and have exponential convergence but require specific conditions, and finite element methods are adaptive but can't hold the conservation law. Even for the basis functions in KANs, splines are good at smooth functions but sensitive to the interval, Chebyshev is faster and more accurate but requires specific interpolation points, Sinc is good at singular functions but can't achieve the same accuracy as Chebyshev on smooth functions.
>
> Even for PINNs which was proposed in 2019, it is not easy to conclude which version of PINNs to use for which type of PDEs. But we believe that SincKAN is a good choice, especially for singular problems.
>
> # Finally, in "vanilla" KANs paper there are some arguments on the advantage of using B-splines, which enable them to use grid-related tricks including grid extension and grid update tricks. In other variants also there are some interesting merits; it's hard to say SincKAN still has all these merits and in addition to them provides more robustness in the case of singularities.
>
> They use grid extension because cubic interpolation is a local interpolation method that requires the interval, but our Sinc, Chebyshev, and wavelet are global interpolation methods thus it is not necessary to update the grids during training. We agree with the statement of [Wav-KAN](https://arxiv.org/abs/2405.12832) that those tricks are limitations instead of merits of vanilla KAN. On the other hand, SincKAN can implement the grid extension more smoothly if we set h to be invariant: recall the interpolation: we can keep the old coefficients and train for the new coefficients as the optimal value is f(jh), which is independent with $N$. Even if $h$ is changed, as $h$ is inversely proportional to  $N$, $h$ will change slightly with $N$ increases, which means the loss won't have an evident jump. To demonstrate it intuitively, consider given $M,N$ with $M>N$ and $ c_j, j=-N,..,N$ is already fully trained, suppose the interpolation before extension is
>
> \begin{equation}
>     \sum_{j=-N}^{N}=c_jS(j,h)
> \end{equation}
>
> After extension,
> \begin{equation}
>     \sum_{j=-M}^{M}=c^\prime_jS(j,h)
> \end{equation}
> we can set $c^\prime_j=c_j \text{ for } j=-N,\cdots,N$, the rest $c_j$ is initialized by zero so that the error after extension is equal to the error before extension.

---

> > ### Comment · Reviewer_FYYo · 2024-11-27
> >
> > Authors addressed my second question fully.
> > The first one remains an open question (and seems authors do agree with me). This makes follow up works on PINNs to be examined with more scrutiny. As reviewers we want to promote works that are as comprehensive as possible. If the method solves a certain class of problems, not an issue but In think it's more suitable for computational journals.

---

> ### Author Response · Authors · 2024-11-25
>
> Thanks for the comment. We understand that the grid extension algorithm doesn't totally convince you, so we conducted experiments on *spectral-basis* and *bl* to show its performance. In Figure 10 and 11, you can find that the evident jump only occurs several times if we use grid extension.
>
> In the experiments, we found that setting a degree = 100 for the total training is always better than increasing the degree from 8 to 100. Besides, because of the cost of changing the basis, increasing the degree is slightly faster than setting degree=100. However, grid extension can be used to find the best number of degrees.  We think it is valuable to discuss this technology in our paper, so we added it in the Appendix, you can find all details in Appendix J.
>
> Look forward to further discussion.

---

### Official Review · Reviewer_mr4i · 2024-11-04

**Soundness:** 3
**Presentation:** 3
**Contribution:** 2
**Rating:** 5
**Confidence:** 2

**Summary:**

This paper presents a new model that applies sinc interpolation to the previously proposed KAN model. It discusses the advantages of the proposed model for function fitting and solving PDEs in a Physics-informed manner.

**Strengths:**

The study explores how the recently developed KAN model can better handle singular problems from the perspective of Physics-Informed Neural Networks (PINNs).

**Weaknesses:**

I believe the main goal of this paper is to improve existing PINN-based models through the development of sincKAN. However, as shown in Table 2, it's unclear if sincKAN demonstrates convincingly better performance than other existing models across various PDE problems. Clearly, as mentioned in the PI-KAN model, there are specific advantages and disadvantages to using KAN for solving PDEs, such as computational cost or training speed. If sincKAN does not consistently outperform across all datasets, as seen in Table 2, what advantages does sincKAN offer over using standard MLP or modified MLP to train Physics-informed Loss? While sincKAN appears to perform better on boundary layer problems, as shown in Table 3, the paper’s title and focus could have been more aligned with boundary layer issues if that is its primary strength. Additionally, are there recent studies related to PINNs specifically aimed at solving boundary layer problems? What would happen if epsilon increased to a larger value, such as 10^8? It might also help to include a basic explanation of KAN’s underlying mechanism for readers unfamiliar with the model, as certain concepts could be difficult to grasp without this context.

**Questions:**

* What exactly is f(jh) in equation (6)? Is it simply the value of f at jh? Is equation (6) interpolating f(x) using those values? If so, do we need to know those values precisely to approximate f(x) with this formulation?
* Could other problems with singularities be tested? Does the proposed method also show significant advantages and accurately approximate solutions in all cases where singular phenomena appear? Could it be applied to other boundary layer problems or higher-dimensional problems beyond two dimensions?
* For line 140, it might be better to use the \paragraph function for terms like “Convergence theorem” or “On a general interval (a, b)” to save space.

---

> ### Author Response · Authors · 2024-11-19
> **For questions**
>
> # What exactly is f(jh) in equation (6)? Is it simply the value of f at jh? Is equation (6) interpolating f(x) using those values? If so, do we need to know those values precisely to approximate f(x) with this formulation?
>
> In equation (6), f(jh) is exactly the value of f at the point x=jh. And yes, equation (6) interprets f(x) using those points. But for approximation problems, it is not necessary to know those values although those points are the theoretical choice if the loss function is exactly equation (6). Let's consider a simple approximation example to further explain it:
> \begin{equation}
>     \min_{\{c_{-N},\cdots,c_N\}} \sum_{i=1}^M \left(f(x_i) - \sum_{j=-N}^N c_j S(j, h)(x_i)\right)^2
> \end{equation}
> where $\{x_1,x_2,\cdots,x_N\}$ are given points. Obviously, if $M=2N+1$, and $x_i=(i-1-N)h$, the problem has an optimal solution $c_j=f(jh)$ because  $S(j, h)(ih)=1  \text{ if } i = j, \text{ otherwise}, S(j, h)(ih)=0.$
> But for other cases, including our case, that is $M\neq 2N+1$ and $x_i$ are randomly sampled, we can also solve this problem by, for example, the least squares method. Herein, we don't need to know those values $f(jh)$ precisely to approximate f(x) with this formulation.
>
> # Could other problems with singularities be tested? Does the proposed method also show significant advantages and accurately approximate solutions in all cases where singular phenomena appear? Could it be applied to other boundary layer problems or higher-dimensional problems beyond two dimensions?
>
> We tested our methods on other problems including fractional PDEs and high-dimensional Poisson problems.
>
> For fractional PDEs, we utilize [fPINN](https://arxiv.org/abs/1811.08967) on the spatial fractional order problem:
> \begin{equation}
>     -\Delta^su+\gamma u=f, \quad \boldsymbol{x}\in (-1,1)^d
> \end{equation}
> if $f=1+\gamma u$, the exact solution is
> \begin{equation}
>     u(x)=\frac{2^{-2s}\Gamma(d/2)}{\Gamma(d/2+s)\Gamma(1+s)}(1-\|x\|^2)^s,
> \end{equation}
> where $s\in \mathbb{R}_+$ is a hyperparameter. In our experiments, $s=0.95,d=1$.
>
> The results are:
> | Model   | Relative error|
> |---------|----------------|
> |  MLP     | 7.32e-02       |
> | SincKAN | 7.28e-03       |
>
> For high-dimensional Poisson equations. we consider
> \begin{equation}
>     \Delta u =f, \quad \boldsymbol{x}\in [0,1]^d.
> \end{equation}
> where $f=2\alpha(2\alpha ||\boldsymbol{x}||^2_2-d) e^{-\alpha ||\boldsymbol{x}||_2^2}$, the exact solution is $u(\boldsymbol{x})=e^{-\alpha ||\boldsymbol{x}||^2_2}$. In our experiments, $d=100, \alpha=10/d$.
>
> The results are:
> | Model   | Relative error|
> |---------|----------------|
> |  MLP     | 7.58e-02      |
> | SincKAN | 3.49e-02     |
>
> # For line 140, it might be better to use the \paragraph function for terms like “Convergence theorem” or “On a general interval (a, b)” to save space.
>
> Thanks for your valuable suggestions! We are considering how to add additional experiments to the main paper.

---

> ### Author Response · Authors · 2024-11-19
>
> # If sincKAN does not consistently outperform across all datasets, as seen in Table 2, what advantages does sincKAN offer over using standard MLP or modified MLP to train Physics-informed Loss? While sincKAN appears to perform better on boundary layer problems, as shown in Table 3, the paper’s title and focus could have been more aligned with boundary layer issues if that is its primary strength.
>
> In our opinion, it is not necessary to be the best model for all datasets. Our work is to explore the implementations of KANs in PINNs and propose a powerful network that is helpful in PINNs. As Sinc interpolation is not as good as Chebyshev and splines when dealing with smooth functions, it is acceptable when we solving smooth PDEs, but SincKAN is not the best. But as the results are good enough, SincKAN excels in handling singular problems. We still believe that SincKAN is a competitive network. We agree that it is a good suggestion that our title should align with our results.
>
> # Are there recent studies related to PINNs specifically aimed at solving boundary layer problems? What would happen if epsilon increased to a larger value, such as $10^8$?
>
> There are several methods proposed to solve boundary layer problems.
>
> 1)  [DAS-PINNs](https://arxiv.org/abs/2112.14038): they use sampling methods to make more points concentrate on the singular part;
> 2) [C-PINN](https://link.springer.com/article/10.1007/s10483-024-3149-8): they embed Chien’s composite expansion method into the loss function;
> 3)  [BL-PINN](https://www.sciencedirect.com/science/article/pii/S0021999122008312): they decompose the boundary-layer problems into two parts and train them individually.
>
> We proposed a novel network that can approximate the boundary layer problems better than MLP. Thus, except for BL-PINN which uses two MLP networks to learn two non-singular parts, we believe SincKAN can be combined with C-PINN and DAS-PINNs as they don't change the network and SincKAN has the better capability of representing singular functions.
>
> We tested SincKAN on $\epsilon=10^8$.  Unfortunately, the performance is poor, $10^8$ is a pretty large number for neural networks.
>
> # It might also help to include a basic explanation of KAN’s underlying mechanism for readers unfamiliar with the model, as certain concepts could be difficult to grasp without this context.
>
> Thanks for your valuable suggestion, we put the explanation of KANs in Appendix, which probably is not clear enough. We will add more explanations on the rebuttal version and mention it in the main paper.

---

> ### Comment · Reviewer_mr4i · 2024-11-21
> **Dear Authors,**
>
> Thank you for your detailed responses to my review comments. While I appreciate the effort you have put into addressing the concerns raised, I find that several critical issues remain unresolved, and your responses have not been reflected in the revision of the paper. This raises questions about why the proposed changes have only been addressed in the rebuttal but not incorporated into the manuscript itself. I provide further clarification of my concerns below
>
> - SincKAN’s Key Strengths and Limitations
>
> You argue that SincKAN does not need to be the best model across all datasets, focusing instead on its strength in solving boundary layer problems. However, this does not sufficiently address why results on datasets where SincKAN does not perform well were included in the first place. If boundary layer problems are the primary strength of SincKAN, the experiments and title should focus more explicitly on this. While you have stated that the title will be revised, no changes have been made to the manuscript, which undermines the credibility of this commitment. Additionally, why were the results included if they do not directly support the central claims of your paper?
>
> - Boundary Layer Problems and ε Scaling
>
> The lack of robust performance for larger values of ε (e.g., ￼) is acknowledged in your response, but no viable alternatives or in-depth analysis of this limitation were provided. Furthermore, your response did not address whether other methods, such as those presented in Finite Element Operator Network for Solving Parametric PDEs (which handled ￼) or Component Fourier Neural Operator for Singularly Perturbed Differential Equations (focused on singular problems), have achieved better performance in similar cases. If boundary layer problems are the central focus of your paper, as implied, comparisons with these studies would strengthen your argument. Including such comparisons and emphasizing singular boundary problem results in the experiments section should be prioritized in a revision.
>
> - Additional Experiments and Explanation
>
> While I appreciate the additional results provided for fractional PDEs and high-dimensional Poisson equations, visualizations (e.g., solution plots) would help to better convey the advantages of SincKAN. This could clarify the contexts where SincKAN excels and where it does not. Could you incorporate these visualizations into the main paper?
> Additionally, the explanation of KAN’s underlying mechanism is still insufficient in the main text, despite earlier feedback highlighting this as a critical issue. Readers unfamiliar with the model may struggle to understand your contributions without a clear and concise explanation included in the manuscript itself.
>
> Therefore, while your responses demonstrate some acknowledgment of the points raised, they have not been adequately incorporated into the paper. The lack of revisions addressing these concerns, particularly those related to experimental focus, visualization, and model explanation, prevents me from reevaluating the contribution of the paper. As such, now, I will stay my initial score.

---

> ### Author Response · Authors · 2024-11-23
>
> Thanks for your further discussion. We planned to submit the final manuscript at the end of this discussion, as our new experiments haven't obtained the error bar. Now we submit the version as you required.
>
> #  The explanation of KAN’s underlying mechanism is still insufficient in the main text
>
> Due to the limitation of the main paper, we think we can only mention it and guide readers to read it in our Appendix. But it is pretty thoughtful and concise for a reader who doesn't know KANs (See G.3).
>
> # Why were the results included if they do not directly support the central claims of your paper?
>
> In Table 1, the approximation experiments including the singular case and smooth cases show that SincKAN has the ability to be the best network. However, due to the limitations discussed in the Conclusion section, SincKAN can't be applied in PINNs for every case. Thus, we want to determine the extent to which those restrictions impair SincKAN's performance in complex but non-single cases.  That is the reason we have results in nonlinear functions, Burgers' equations, and Navier-Stokes equations: we want to show that although SincKAN has those limitations, SincKAN is still good enough for general cases. Thus, we agree with changing the title since approximation problems are not very practical and we pay more attention on the PINNs.
>
> # Boundary Layer Problems
> 1) The [FEONet](https://arxiv.org/abs/2308.04690) is a hybrid framework that combines neural networks and traditional methods. Such a framework is always better than pure PINNs because it inherits the accuracy of traditional methods. But there are several drawbacks including complexity and grid generation. The most significant drawback is that it can't solve high-dimensional problems which PINNs are famous for, because the basis functions of finite element methods have to decompose the whole domain so that the complexity increases exponentially with the dimension. Herein FEONet also inherits the drawback of suffering the curse of dimensionality. So we don't need to compare with this paper.
>
> 2) The [ComFEO](https://arxiv.org/abs/2409.04779) solving boundary layer problems by Neural operators which are usually supervised learning  while PINNs are unsupervised learning. This paper generates solutions from traditional numerical methods (see the second paragraph of Experiments in ComFEO). So we don't need to compare with this paper.
>
> For the papers we mentioned above, the largest $\epsilon$ is $10^4$, and the experiments we did on it show that SincKAN can obtain $5.27e-03$\pm$1.29e-03$ which is also good enough, we added this experiments on our paper. Besides, we think the performance can be better if SincKAN is equipped with these mentioned approaches.
>
> #  Could you incorporate these visualizations into the main paper?
>
> Yes, we plot the errors and training loss of fractional PDE into the main paper (Figure 4)and plot the solutions in the Appendix (Figure 7).
>
> And for high-dimensional Poisson equations, as it is not practical to plot the solution, so we plot the relative error of every point. Our testing points are composed of 80% interior points and 20% boundary points. To show our points are in a reasonable distribution, we also plot the location in a 2D profile (Figure 5).

---

> ### Author Response · Authors · 2024-11-28
>
> Dear reviewer,
>
> We are sorry about the lack of revision, that make you stay the score. We believe that we have already answered all questions and enriched the manuscript by tables, figures and explanation. Is there anything else that we haven't answered or haven't answered fully? We are glad to have further discussion with you or get a positive feedback from you.

---

### Author Response · Authors · 2024-11-23

Many reviewers doubted that :

# 1) the relationship between Theorem 1 and SincKAN.

Theorem 1 and 2 are for Sinc numerical methods, especially the Sinc interpolations. Those two theorems inspired us to use Sinc interpolation in KANs. However, because directly applying the Sinc interpolation is not a good idea as we discuss in our manuscript, we unfold this approach to meet the demands of machine learning. Thus, this theorem is merely our fundamental theorem for replacing the cubic interpolation in KANs. Furthermore, interpolation is one of the most important compositions in KANs but not all of KANs. Thus, Theorem 1 and 2 can't dominate SincKAN totally.

# 2) SincKAN is not the best in every experiment.
Our research explores the implementations of KANs in PINNs and proposes a powerful network that is helpful in PINNs. As Sinc interpolation is not as good as Chebyshev and splines when dealing with smooth functions, it is acceptable that SincKAN is not the best when solving smooth PDEs. However, because SincKAN excels in handling singular problems and the results of smooth cases are good enough, we think SincKAN is a competitive network for PINNs.

Additionally, some reviewers argued that why we put the experiments that SincKAN is not the best. In Table 1, the approximation experiments including the singular case and smooth cases show that SincKAN has the ability to be the best network. However, due to the limitations discussed in the Conclusion section, SincKAN can't be applied in PINNs for every case. Thus, we want to determine the extent to which those restrictions impair SincKAN's performance in complex but non-single cases.  That is the reason we have results in nonlinear functions, Burgers' equations, and Navier-Stokes equations: we want to show that although SincKAN has those limitations, SincKAN is still good enough for general cases.

# 3) the performance of SincKAN on more complex problems.
We added two extra experiments to show the performance on complex problems:
## Fractional PDEs
We consider the fractional spacial order problem:
\begin{equation}
    -\Delta^su+\gamma u=f, \quad \boldsymbol{x}\in (-1,1)^d
\end{equation}
if $f=1+\gamma u$, the exact solution is
\begin{equation}
    u(x)=\frac{2^{-2s}\Gamma(d/2)}{\Gamma(d/2+s)\Gamma(1+s)}(1-\|x\|^2)^s,
\end{equation}
where $s\in \mathbb{R}_+$ is a hyperparameter. We consider $d=1$ case, the results are:
|model|$s=0.85$|$s=0.95$|
|----|----|---|
MLP  | 8.84e-03  | 7.32e-02 |
SincKAN | 6.57e-04 |7.28e-03 |

## Poisson equations
The second one is high-dimensional Poisson equations:

\begin{equation}
    \Delta u =f, \quad \boldsymbol{x}\in [0,1]^d.
\end{equation}
where $f=2\alpha(2\alpha ||\boldsymbol{x}||^2_2-d) e^{-\alpha ||\boldsymbol{x}||_2^2}$, the exact solution is $u(\boldsymbol{x})=e^{-\alpha ||\boldsymbol{x}||^2_2}$. In our experiments, $d=100$, $\alpha=10/d$. The results are:
| Model   | Relative error|
|---------|----------------|
|  MLP     | 7.58e-02      |
| SincKAN | 3.49e-02     |

The main changes in our paper are:

1) add $\epsilon=10^4$ boundary layer problems, fractional PDEs, and high-dimensional Poisson as extra experiments.

2) added a theoretical proof of coordinate transformation.

3) added a thoughtful introduction to KAN.

4) changed the title to more focus on singular problems.

5) added a discussion of the grid extension.

6) added more experiments on approximation.

---

### Author Response · Authors · 2024-11-24

Dear AC,

We kindly ask to additionally encourage reviewer wJoX interact with us during the discussion period. It is our belief that we addressed concerns of wJoX and we would like to receive additional comments or reevaluation of the score.

---

### Meta-Review · Area_Chair_XFg2 · 2024-12-17

**Metareview:**

This paper proposes the use of sinc interpolation in Kolmogorov-Arnold networks (KANs) with the goal of achieving better function interpolation and for the solution of PDEs in the context of physics-informed neural networks (PINNs). I thank the authors and the reviewers for their discussions. From these, I believe improving KANs is an interesting research direction, in particular, within the context of solving PDEs in PINNs.

The major criticism of this paper is in terms of the experimental evaluation, with regards to, e.g., the focus of the paper (should it be on boundary layer problems?), the lack of robustness, more visualizations, better explanations of the KANs underlying mechanism, and the unsuitability of the benchmarks problems used for evaluation.

I strongly disagree with the belief of some reviewers (Reviewer mr4i, other reviewers at ICLR and in other top ML venues) that an ML paper must be shown to be the best across all datasets and (worse) that those problem where the method does not do well should not be reported. In fact, as a community, we need to commend authors to do otherwise. However, I believe the point on the unsuitability of the benchmarks used is an important one. Although the authors have  added new results, some of the reviewers remained unconvinced. Because of this and the additional issues outlined above, I recommend rejection.

**Additional Comments On Reviewer Discussion:**

The major criticism of this paper is in terms of the experimental evaluation, with regards to, e.g., the focus of the paper (should it be on boundary layer problems?), the lack of robustness, more visualizations, better explanations of the KANs underlying mechanism, and the unsuitability of the benchmarks problems used for evaluation.

I strongly disagree with the belief of some reviewers (Reviewer mr4i, other reviewers at ICLR and in other top ML venues) that an ML paper must be shown to be the best across all datasets and (worse) that those problem where the method does not do well should not be reported. In fact, as a community, we need to commend authors to do otherwise. However, I believe the point on the unsuitability of the benchmarks used is an important one. Although the authors have  added new results, some of the reviewers remained unconvinced. Because of this and the additional issues outlined above, I recommend rejection.

---

### Decision · Program_Chairs · 2025-01-22

Reject